# QTL mapping of human retina DNA methylation identifies 87 gene-epigenome interactions in age-related macular degeneration

Jayshree Advani[1], Puja A. Mehta [2,3,4], Andrew R. Hamel [2,3,4], Sudeep Mehrotra[2,3,4], Christina Kiel [5], Tobias Strunz[5], Ximena Corso-Díaz[1], Madeline Kwicklis[1], Freekje van Asten[1], Rinki Ratnapriya[1], Emily Y. Chew [6], Dena G. Hernandez[7], Sandra R. Montezuma[8], Deborah A. Ferrington[8,10], Bernhard H. F. Weber [5,9], Ayellet V. Segrè[2,3,4] ✉ & Anand Swaroop [1] ✉

DNA methylation provides a crucial epigenetic mark linking genetic variations to environmental influence. We have analyzed array-based DNA methylation profiles of 160 human retinas with co-measured RNA-seq and >8 million genetic variants, uncovering sites of genetic regulation in *cis* (37,453 methylation quantitative trait loci and 12,505 expression quantitative trait loci) and 13,747 DNA methylation loci affecting gene expression, with over one-third specific to the retina. Methylation and expression quantitative trait loci show non-random distribution and enrichment of biological processes related to synapse, mitochondria, and catabolism. Summary data-based Mendelian randomization and colocalization analyses identify 87 target genes where methylation and gene-expression changes likely mediate the genotype effect on age-related macular degeneration. Integrated pathway analysis reveals epigenetic regulation of immune response and metabolism including the glutathione pathway and glycolysis. Our study thus defines key roles of genetic variations driving methylation changes, prioritizes epigenetic control of gene expression, and suggests frameworks for regulation of macular degeneration pathology by genotype–environment interaction in retina.

Common healthy and disease traits in humans exhibit extensive variability, are largely multifactorial, and dictated by a complex interplay between genetic architecture and widely varying environments[1]. Genomic variations can impact phenotypes through genetic and epigenetic control of gene expression programs. Large genome-wide association studies (GWAS) have been effective in identifying thousands of genetic variants that are linked to common traits (https://www.ebi.ac.uk/gwas/); however, a vast majority of the associated variations are present in non-coding regions of the genome, likely

impacting gene regulation and consequently disease pathogenesis[2–4]. The GTEx project has provided extensive descriptions of expression quantitative trait loci (eQTLs) for many human tissues[5,6]. Nonetheless, an array of dynamic environmental factors, including diet and socioeconomic status, can complicate the interpretation of GWAS datasets[7]. Furthermore, many common traits are also influenced by advanced age, and genetic inheritance and environment are critical determinants of the aging process itself[8,9]. GWAS success has so far been limited in unraveling the complexities of gene-environment relationships and

the contribution of individual genetic variations to multifactorial phenotypes.

DNA methylation (DNAm) is a key dynamic epigenetic mark that is established during development and largely preserved in differentiated cell types to maintain chromatin organization and genetic controls[10–12]. DNAm is an active contributor to gene regulation, and tissue-specific aberrations in DNAm landscape are associated with environmental factors (such as diet and exercise), aging as well as age-related diseases[11–14]. Non-coding variants, both in *cis* and *trans*, can exert strong influence on DNAm, alter chromatin topology, and modulate the expression of target genes[15–18]. Integrated analyses of genetic variants affecting DNAm (mQTLs), association between DNAm sites and gene expression (eQTMs), and GWAS of complex traits have only recently begun to elucidate the complex relationships among multiple disease-causing factors[19–21]. However, no such information currently exists for eye or retinal tissues and traits.

Age-related macular degeneration (AMD) is a multifactorial progressive neurodegenerative disease, which is characterized by loss of central vision and is the leading cause of irreversible vision impairment and blindness in older individuals worldwide[22]. Patients with AMD exhibit lipid-rich extracellular deposits (drusen), atrophy of the retinal pigment epithelium (RPE), and loss of photoreceptors primarily in the central macular region of the retina. AMD pathology begins in the macula; however, it is a pan-retinal disease with early lesions detected in the peripheral retina as well[23–25]. Advanced age is arguably a major critical component in addition to genetic susceptibility and environmental factors (such as smoking and diet), which together determine etiology and varying phenotypes of AMD[22]. A large AMD GWAS had previously identified association of 52 independent genetic variants at 34 loci[26], which have been expanded to 46 loci in a larger GWAS meta-analysis[27] and 63 loci by a cross-ancestry GWAS[28]. Causal genes and functionally relevant variants at most AMD-associated loci are still unrecognized, though a few key biological pathways have begun to emerge[29]. Ultra-rare variants in case-control or family-based genetic studies can potentially point to causal genes[26,30], as exemplified by the identification of complement 8 A and C8B[31]. Furthermore, like other complex traits, a majority of AMD-associated variants are present in non-coding regions that could regulate expression and epigenetic landscape of distal genes. Integrated statistical analyses of GWAS with gene expression quantitative trait loci (eQTLs) in retina and GTEx tissues have helped in prioritizing potential target genes for AMD[32–35]. High-resolution mapping of human retinal genome topology further helps elucidate chromatin looping patterns of variants in distal *cis*-regulatory elements, such as enhancers, and refines candidate disease-causing genes[36]. Despite innovative advances, we have limited understanding of underlying mechanisms that associate genetic regulation with epigenomic shifts linked to aging and environmental factors in AMD pathogenesis.

QTL mapping of DNAm and its integration with GWAS and eQTLs in the human retina can potentially uncover epigenomic regulation of disease pathogenesis, as demonstrated for multiple other tissues[20,21,37]. We note that alterations in DNAm are also associated with aging in the retina[38,39]. Here, we generated genome-wide DNAm profiles of 160 human retina samples to identify associations between genetic, epigenetic, and transcriptional variation relevant to retinal homeostasis and complex disease traits, such as AMD. We report mapping of mQTLs and eQTMs and an integrative analysis of mQTLs, eQTLs, and AMD-GWAS variants. Using Summary data-based Mendelian Randomization (SMR), multiple colocalization methods, and Hi-C data, we demonstrate complex associations among genetic variants, DNAm, and gene expression and identify 87 unique genes affected by DNAm that may contribute to AMD risk. Our studies provide insights into molecular mechanisms underlying epigenomic regulation of AMD and suggest aging and environment-responsive pathways.

## Results

### Overview of the analysis workflow

We designed an integrative analysis workflow of multiple human retina omics datasets, incorporating DNAm, gene expression, imputed genotypes, AMD-GWAS, and Hi-C data (Fig. 1a). We carried out DNAm profiling of postmortem retinas (n = 160), with an equal distribution of males and females and mean age of 73 years, using the Human MethylationEPIC BeadChip (Supplementary Data 1). After quality control (QC) and covariate analysis, including adjusting for AMD grade (see Methods) (Supplementary Fig. 1a–d), DNAm data from 152 retina samples was integrated with the previously published corresponding genotype and RNA-seq data[32,34] for *cis*-mQTL mapping, and association of DNAm of CpG sites with gene expression (*cis*-eQTMs). We also performed concurrent *cis*-eQTL analysis from 403 retinas with genotype and RNA-seq data from the same study[32] correcting for covariates (see Methods). To provide a comprehensive set of causal relationships between *cis*-mQTLs and *cis*-eQTLs, and between m/eQTLs and AMD GWAS signals, we pursued two complementary approaches: (i) SMR to distinguish pleiotropic or causal association from linkage of genetic associations with DNAm, gene expression and AMD[19,40,41], and (ii) colocalization analyses that test whether co-occurring association signals are tagging the same causal variant/haplotype, including eCAVIAR[42] (assuming two causal variants), coloc[43] (assuming a single causal variant), and multiple-trait-coloc (moloc; assuming up to four causal variants)[44]. We also integrated adult retina Hi-C data including chromatin loops, and *cis*-regulatory elements (CREs) and super-enhancers (SEs)[36] inferred from chromatin histone marks with mQTLs, eQTLs, eQTMs, and SMR or moloc associations, to identify high confidence candidate AMD and QTL target genes through physical linking between variants, CpG sites and genes.

### The landscape of retina mQTLs and eQTLs

To characterize genetic regulation of DNAm in human retina, we performed *cis*-mQTL (n = 152) analysis on all genotyped and imputed variants in *cis* (±1 Mb) of 749,158 CpG sites that passed stringent QC criteria (see Methods) using QTLtools[45,46]. We controlled for genetic population structure with top 10 genotype principal components (PCs), sex, age, collection site and other hidden confounding effects with inferred surrogate variables (SVs), and AMD grade even though no significant differential CpG methylation was observed between AMD grades (see Methods). We identified 2,817,314 significant variant-CpG *cis*-mQTLs (False discovery rate (FDR) ≤ 0.05) for 36,906 CpG sites that map to 10,000 mGenes (Methods, and Supplementary Data 2). We detected 37,453 independent mQTL signals using conditional analysis (Supplementary Fig. 2a), with 98.5% having a single independent signal per CpG site (Supplementary Data 2, Supplementary Fig. 2c). Only 564 (1.5%) CpG sites had two independent mQTL signals. Concurrent analysis of all variants within ±1 Mb of 17,382 genes expressed in the retina (n = 403, Methods) revealed 2,023,293 significant variant-gene *cis*-eQTLs (FDR ≤ 0.05) for 9395 eGenes (eQTL target genes); of these, 12,505 are independent eQTLs (Supplementary Fig. 2b). A large fraction of eGenes (26.8%) showed more than one independent genetic effect in contrast to the CpG sites, though more independent signals for CpG methylation might be detected with a larger sample size (Supplementary Data 3, Supplementary Fig. 2c). Almost 95% of the mQTL variants are clustered near CpGs (median distance of 11.4 kb) (Supplementary Fig. 2d), as recently identified for other tissues[21], whereas 95% of eQTL variants clustered near the transcriptional start site (TSS) of the corresponding eGene (median distance of 5.5 kb) (Supplementary Fig. 2e). A majority of CpGs (including all tested and those with significant mQTLs) are present within 200 bp or 1500 bp of TSSs, 5′ UTR, gene body and intergenic regions. In addition, CpGs with significant mQTLs were relatively depleted from exon boundary and 3′ UTR regions when compared to all CpGs (Fig. 1b).

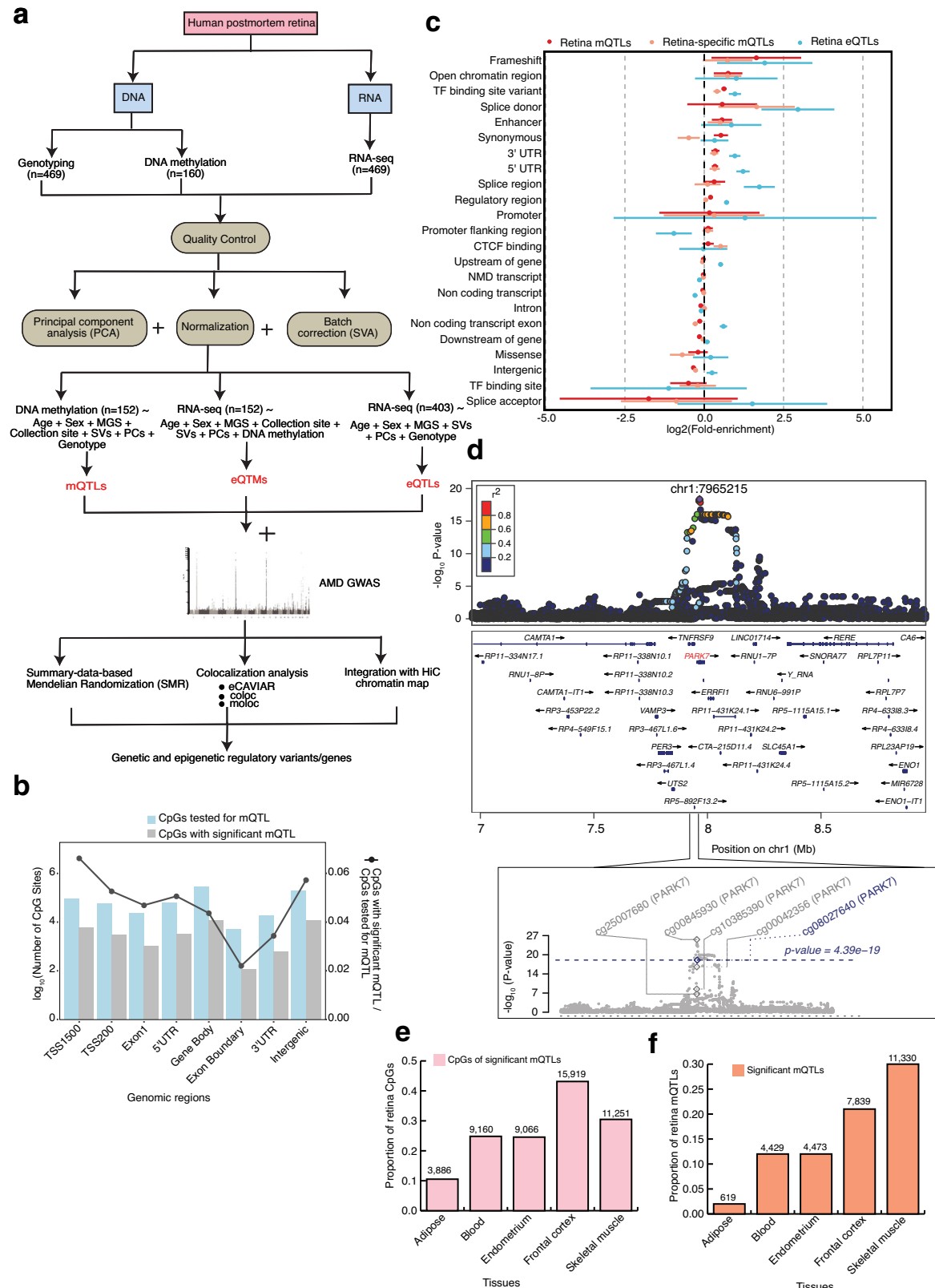

## Genomic features and biological processes enriched in retina mQTLs compared to eQTLs

We examined the enrichment of mQTL variants (FDR ≤ 0.05) in functional genomic elements and compared the results to eQTL variants using TORUS[47,48] (see Methods). Retina mQTLs were significantly enriched in both coding regions and transcriptional regulatory elements, with the strongest enrichment detected in frameshift variants, followed by open chromatin regions, transcription factor (TF) binding sites, enhancers, synonymous variants, and 3' and 5' UTR (Fig. 1c). Though highly enriched among frameshift variants, this class of mQTLs accounted for only 214 genetic variants associated with CpG methylation (mVariants) (Supplementary Fig. 2f, right panel).

**Fig. 1 | Graphic summary of datasets generated, integrated and analyses performed in the present study and robust identification of retina mQTLs.**
**a** Schematic representation of our genetic, epigenetic, and transcriptomic datasets and methods used in the identification and integration of methylation quantitative trait loci (*cis*-mQTL), expression quantitative trait loci (*cis*-eQTL) and expression quantitative trait methylation (*cis*-eQTM) with AMD GWAS and retina Hi-C chromatin map. **b** Number of CpG sites tested (blue) and significant (grey) in various genomic regions in mQTL analysis. **c** QTL enrichment in functional annotations for all retina (red) or retina-specific (orange) *cis*-mQTLs identified from n = 152 biologically independent samples and all retina *cis*-eQTLs identified from n = 403 biologically independent samples (blue). Points (centre) refer to m/eQTL fold-enrichment estimates on log2 scale with 95% confidence intervals (lines), shown in descending order based on the retina mQTL fold-enrichment across annotations with >550 variants per QTL type. **d** LocusZoom plot showing the retina mQTL association, $-\log_{10}(P$ value) for the top mQTL signal with CpG (cg08027640) with $P$ value = 4.39 × 10$^{-19}$ for *PARK7* gene. The diamond indicates the top mVariant (chr1:7965215:C:T; rs7517357) for the independent cg08027640 mQTL signal. The color of the points is determined by their linkage disequilibrium (LD) with respect to the lead mVariant in purple. The bottom plot shows $-\log_{10}(P$ values) of the variant association with five different CpGs in *PARK7* gene region from mQTL results. The grey and blue diamond's represent $-\log_{10}(P$ values) of the lead mVariants for four CpGs and cg08027640, respectively. **e** Proportion of retina CpGs with significant mQTLs that are also significant mQTLs across different tissues. **f** Proportion of retina mQTLs that are also significant mQTLs across different tissues.

Synonymous variants, enhancers, and open chromatin regions accounted for the largest number of enriched mQTLs (thousands of mVariants; Supplementary Fig. 2f, right panel). In contrast, eQTLs were strongly enriched among variants that affect splicing, followed by frameshift variants, 5' UTR, TF binding sites, 3' UTR, and regulatory regions (Supplementary Fig. 2f, left panel). Though not significantly enriched in open chromatin regions and enhancers (lower bound 95% confidence interval <0), the fold-enrichment of eQTLs in these functional categories was higher than that of mQTLs.

We next examined the pathways that are enriched for genes potentially regulated by mQTLs (mGenes) in the retina using gene ontology (GO) enrichment analysis (see Methods). The mGenes are enriched (FDR ≤ 0.05) in a range of biological processes, including those related to cell adhesion, actin filament organization, synaptic signaling, and peptide hormone secretion (Supplementary Fig. 3a, and Supplementary Data 4). Examples of genes driving these GO enrichments include *PARK7*, a mitochondrial gene involved in synaptic signaling and a potential target of 5 mQTLs (Fig. 1d), and *MTOR* involved in the actin filament-based process, regulation of GTPase activity, control of cell growth and proliferation and a potential target of 2 mQTLs (Supplementary Fig. 3b and Supplementary Data 4). In contrast, target genes of retina eQTLs (eGenes) are enriched in cellular components (FDR ≤ 0.05), such as extracellular matrix and endoplasmic reticulum lumen (Supplementary Fig. 3c, and Supplementary Data 5). Thus, genetic effects on CpG methylation in the retina appear to be driven by distinct molecular mechanisms and biological processes compared to the genetic effects on gene expression.

## Tissue-specificity of retina mQTLs
To evaluate the tissue-specificity of DNAm and mQTLs, we compared 36,906 methylated CpGs with significant mQTLs and 37,453 independent mQTLs in the retina to those identified in five different tissues with available mQTLs: adipose (n = 119)[49], blood (n = 614)[50], endometrium (n = 66)[51], brain frontal cortex (n = 526)[52], and skeletal muscle (n = 282)[20]. We detected the highest representation of retina methylated CpGs in the frontal cortex (43%), followed by skeletal muscle (30%), blood (24%), endometrium (24%), and adipose (10%) (Fig. 1e). Our results reflect the shared neuronal nature of frontal cortex and retina. The retina mQTLs also show the highest overlap (22–30%) with mQTLs in skeletal muscle and frontal cortex (Fig. 1f). Interestingly, 13,458 mQTLs and 5895 CpGs are unique to the retina (Supplementary Data 2, 6). Retina-specific methylated CpGs are similarly distributed within and around the gene body, and least represented in exon boundaries and 3' UTR, as with all the methylated CpGs identified in the retina (Supplementary Fig. 3d). Retina-specific mQTLs are also enriched in similar functional genomic elements as all retina mQTLs, aside for showing significantly stronger enrichment in splice donor variants compared to all retina mQTLs, as in case of retina eQTLs (Fig. 1c). Thus, retina-specific mQTLs may be enriched for genetic effects on alternative isoform expression compared to all retina mQTLs. Gene ontology analysis of the retina-specific mGenes reveals an enrichment (FDR ≤ 0.05) in unique biological processes that include synaptogenesis and photoreceptor cell maintenance (Supplementary Fig. 3e, and Supplementary Data 7).

## Identification of eQTMs in the human retina
We evaluated the association between DNAm of CpG sites in *cis* (±1 Mb) and genes expressed in the retina, considering 732,506 CpG sites and 18,263 genes. Top 10 genotype PCs, known covariates (e.g., AMD grade), and SVs capturing hidden covariates of expression, were included in the linear regression model (see Methods). We identified a total of 13,747 significant *cis*-eQTMs (FDR ≤ 0.05) in the retina, comprised of 10,585 unique CpGs (1.4% of tested CpGs) regulating 13,747 unique genes (75.2% of total genes); of which, 11,248 (82%) are protein-coding genes (Supplementary Data 8). Of the 13,747 eQTMs, 770 CpGs and 7292 genes showed a significant mQTL and eQTL, respectively. All eQTMs were independent signals, and none had a secondary signal with our current sample size. Most CpGs with significant eQTMs resided proximal to the target gene's TSS with a median distance of 1.07 kb (Fig. 2a) and are most enriched (81%) within 1500 bp and 200 bp of TSSs, first exon, 5' UTR, and the gene body (Fig. 2b), similar to the CpGs with significant mQTLs. The eQTM target genes are enriched (FDR < 0.05) in mitochondrial and translation-related processes (Supplementary Fig. 4a, and Supplementary Data 9).

We further examined the direction of effect of CpG methylation on gene expression. A higher fraction of CpGs showed a canonical negative correlation with their target gene expression (54.5%) compared to a positive correlation (45.5%), as observed in other tissues[20,21,37] (Supplementary Data 8). For example, the CpG cg24846343 located in a gene body is negatively correlated with expression levels of Glutathione S-transferase, *GSTT2B* (Fig. 2c), and CpGs cg21653793 and cg10832655 located in a 5' UTR region are negatively correlated with the expression of cholesterol transporter *ABCA1* and the neuron derived neurotrophic factor *NDNF*, respectively (Fig. 2d, and Supplementary Fig. 4b). On the contrary, CpG cg24307499 located in a gene body is positively correlated with the expression of *NLRP2*, an immune response regulator, and CpG cg04718426 located within 200 bp of the TSS of the zinc finger protein, *ZNF232* is positively correlated with the expression of *ZNF232* (Fig. 2e and Supplementary Fig. 4c). We took chromatin accessibility (ATAC-Seq) data from our previously published study on adult human retina[36] that recorded open chromatin regions observed in at least 3 out of the 5 samples. We examined chromatin accessibility footprints for 13,747 significant eQTMs of which 6267 target genes (45%) are positively correlated with CpG methylation, and 7480 genes (55%) are negatively correlated with CpG methylation. Of these, we identified 5057 genes (80.6% of the target genes) of positively correlated eQTMs and 5974 genes (79.8% of the target genes) of negatively correlated eQTMs overlapping an open chromatin region in the retina.

Next, we inspected the distribution of number (fraction) of eQTMs with their target and all known genes across different chromosomes and uncovered a greater relative number of eQTMs on a few smaller chromosomes, e.g., chromosome 16, 17, and 19 (Fig. 2f). While on average eQTMs regulate a single gene, we noticed that 0.2%

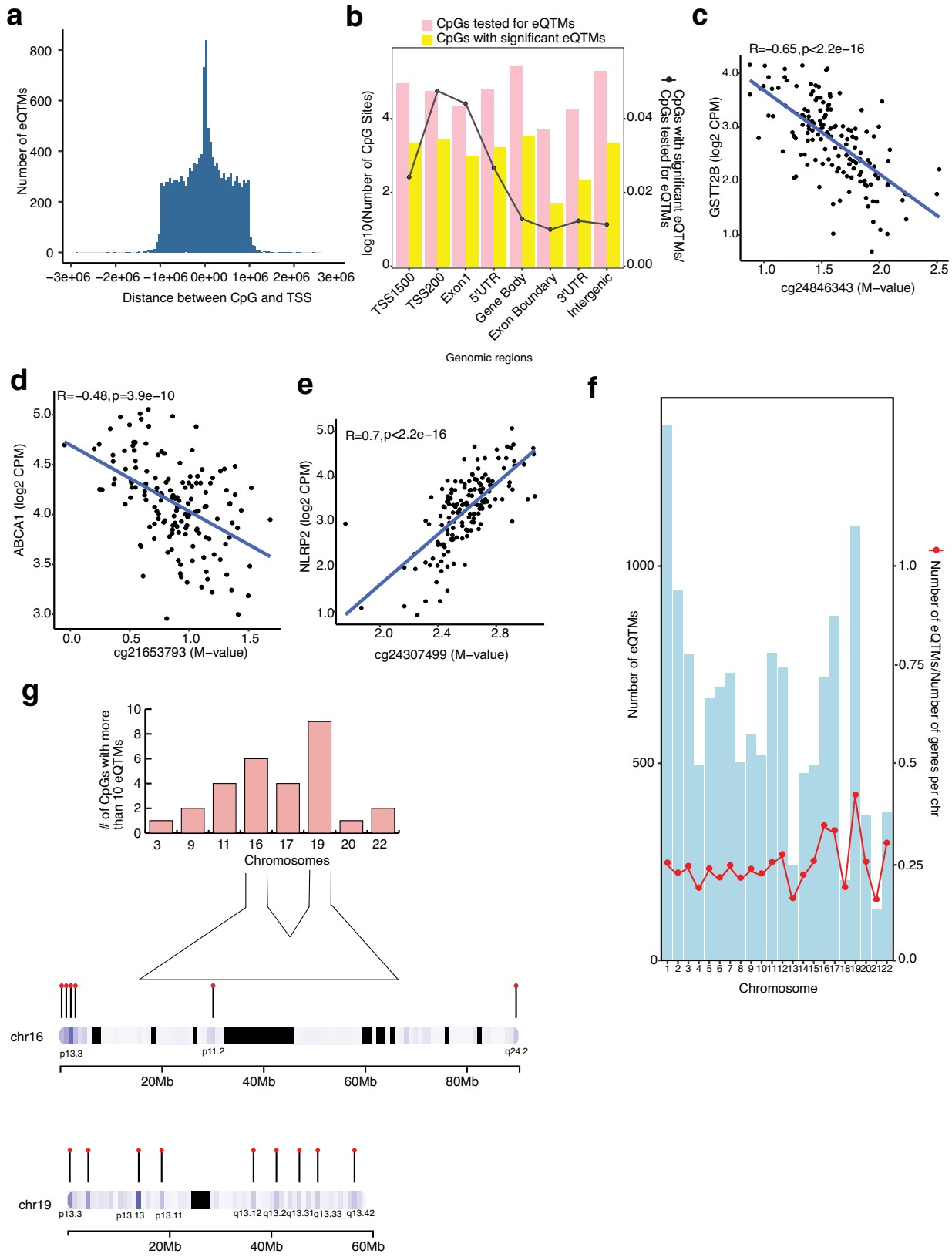

of CpGs regulated more than 10 genes in the retina and that a majority of these are on chromosomes 16 and 19 (Fig. 2g). Gene ontology analysis of these genes revealed an enrichment (FDR ≤ 0.05) of catabolic processes, including autophagic mechanisms and proteasomal protein degradation (Supplementary Fig. 4d and Supplementary Data 10). The CpGs that regulate more than 10 genes and

thereby have greater pleiotropic effects are clustered on p and q arms of both chromosomes 16 and 19 (Fig. 2g). Genes on chromosome 16 are enriched in regulation of telomere maintenance (Supplementary Fig. 4e and Supplementary Data 11) and on chromosome 19 in negative regulation of transcription by RNA polymerase II, RNA splicing via transesterification reactions, and positive regulation of

**Fig. 2 | Characterization and distribution of retina eQTMs. a** Distribution of the distance between the CpG and the transcription start site (TSS) of the respective gene is plotted against the number of eQTMs. **b** Combination chart representing the number of CpG sites tested (pink) and significant (yellow) in various genomic regions in eQTM analysis. **c, d, e** DNAm levels are presented on the X-axis and the normalized gene expression levels are shown on the Y-axis. Pearson's correlation coefficient (R) was calculated between methylation and gene expression. **c** eQTM for CpG cg24846343 located in gene body and *GSTT2B* on chromosome 22 with

R = −0.65, $p < 2.2 \times 10^{-16}$. **d** eQTM for CpG cg21653793 located in 5'UTR and *ABCA1* on chromosome 9 with R = −0.48, $p = 3.9 \times 10^{-10}$. **e** eQTM for CpG cg24307499 located in gene body and *NLRP2* on chromosome 19 with R = 0.7, $p < 2.2 \times 10^{-16}$. **f** Distribution of number of eQTMs on different chromosomes and eQTM fraction (red points) relative to the total number of genes per chromosome. **g** Top panel: Number of CpGs that regulate more than 10 eQTMs and are distributed on various chromosomes. Bottom panel: Cluster of CpGs on chromosomes 16 and 19 on arm p and q.

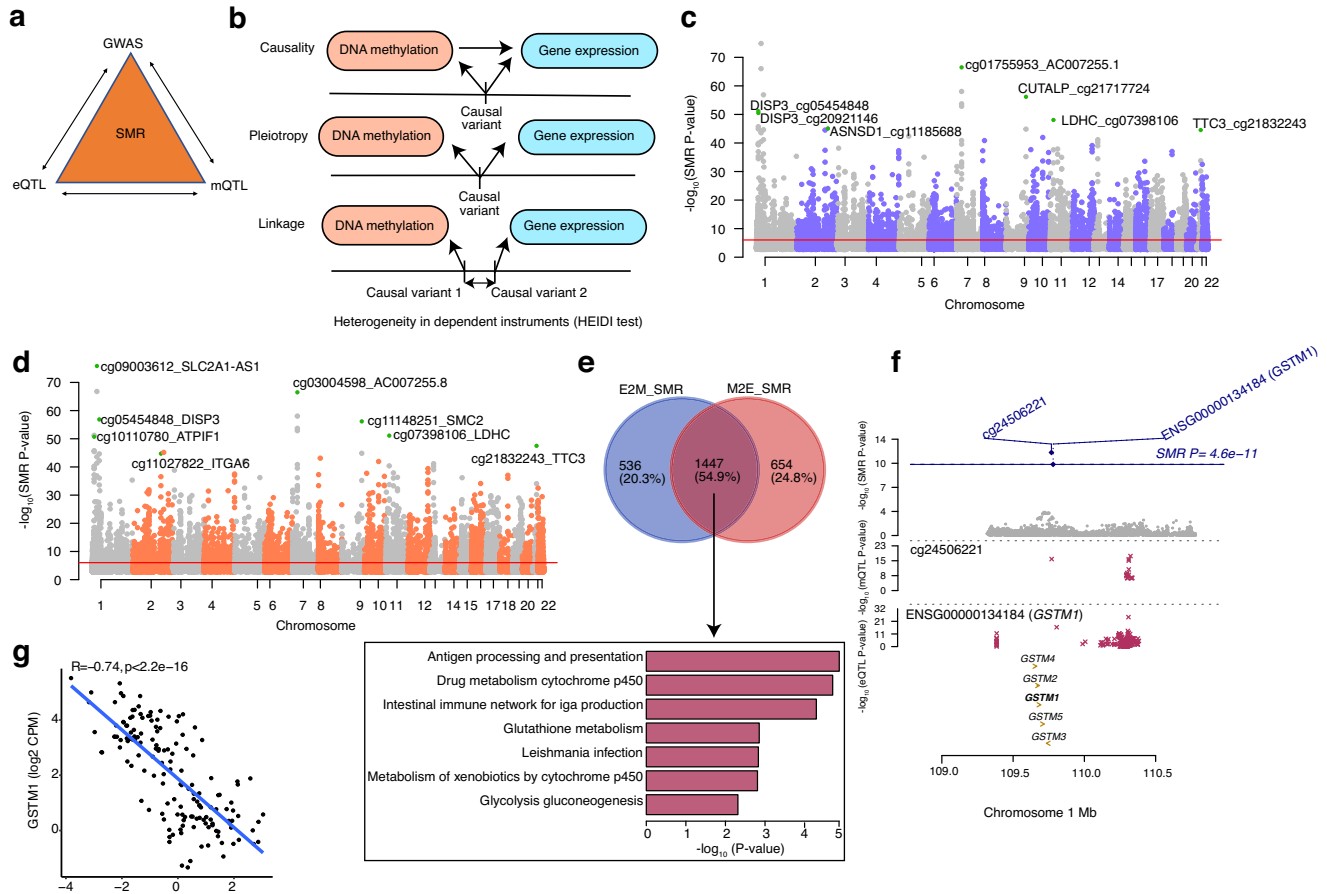

**Fig. 3 | Associations between retina DNA methylation and gene expression through genotypes. a** Schematic of bidirectional integrative analysis that integrates summary-level data from independent GWAS with data from retina mQTL and eQTL. **b** Heterogeneity in dependent instruments (HEIDI) model to distinguish pleiotropy from linkage for an observed association between DNAm and gene expression through genotypes. **c** Manhattan plot of SMR tests for association between gene expression and DNAm (E2M_SMR). Shown on the y axis are the −log₁₀(P value) from SMR tests. The red horizontal lines represent the genome-wide significance level (SMR P value = $9.29 \times 10^{-7}$). **d** Manhattan plot of SMR tests for association between DNAm and gene expression (M2E_SMR). Shown on the y axis is the −log₁₀(P value) from SMR tests. The red horizontal lines represent the genome-wide significance level (SMR P value = $9.38 \times 10^{-7}$). **e** Venn-diagram representing common and unique genes identified in E2M_SMR and M2E_SMR associations and bar graph representing the enriched pathways identified in the

pathway analysis of common genes at FDR < 0.05. The y axis shows the −log₁₀(Empirical P value) from *GeneEnrich*. **f** Results of variants and SMR associations across DNAm and gene expression (M2E_SMR) in the *GSTM1* locus on chromosome 1. The top plot shows −log₁₀(SMR P values) of SNPs from the SMR analysis of DNAm and gene expression (M2E_SMR). The blue diamonds represent −log₁₀(SMR P value) from SMR tests for associations of DNAm and *GSTM1* expression with SMR P value = $4.6 \times 10^{-11}$. The second plot shows −log₁₀(P values) of the SNP association for DNAm probe cg24506221 from the mQTL data. The third plot shows −log₁₀(P values) of the SNP associations for gene expression of *GSTM1* from the eQTL data. **g** eQTM for CpG cg24506221 located in TSS200 region and *GSTM1* on chromosome 1. DNAm levels of cg24506221 are presented on the X-axis and the normalized gene expression levels are shown on the Y-axis. Pearson's correlation coefficient was calculated between methylation and gene expression with R = −0.74, $p < 2.2 \times 10^{-16}$.

cellular protein catabolic process (Supplementary Fig. 4f and Supplementary Data 12).

### Causal or pleiotropic relationships between genetic regulation of DNAm and gene expression

Variants that regulate CpG methylation (mQTLs) may in turn affect gene expression, and variants that regulate gene expression (eQTLs)

may influence CpG methylation; however, the extent to which these molecular mechanisms occur in the retina is not clear. We thus used Summary-data-based Mendelian Randomization (SMR)[40] to examine whether mQTLs underlie the causal mechanism or share the same causal variant (pleiotropy) with eQTLs in the retina and vice versa (Fig. 3a). We performed SMR analysis with retina eQTL and mQTL summary statistics considering DNAm as the exposure and gene

expression as the outcome (represented as M2E_SMR) or gene expression as the exposure and DNAm as the outcome (represented as E2M_SMR). We further applied a heterogeneity in dependent instruments (HEIDI) test to differentiate a causal or pleiotropic model from a linkage model (Fig. 3b). In the E2M_SMR analysis, we identified 7869 associations (SMR $P$ value $< 9.30 \times 10^{-7}$ after Bonferroni correction) (Supplementary Data 13); of these, 5805 associations passed the linkage test (HEIDI $P$ value $> 0.05$), corresponding to 2256 eQTLs, 5612 mQTLs and 1983 genes (Fig. 3c). In the M2E_SMR analysis, 9307 associations (SMR $P$ value $< 9.38 \times 10^{-7}$ after Bonferroni correction) (Supplementary Data 14) were evident, of which 6592 associations passed the linkage test (HEIDI $P$ value $> 0.05$) corresponding to 5175 mQTLs, 5012 eQTLs and 2101 genes (Fig. 3d; 51% of the M2E_SMR QTL effect sizes were negatively correlated and 49% positively correlated). Of the 6592 M2E_SMR associations, 232 associations (3.5%) corresponding to 232 genes were also identified as significant eQTMs, of which 176 (75.8%) eQTMs showed negative correlation and 56 (24.1%) eQTMs revealed positive correlation. We detected 554 common associations between E2M_SMR and M2E_SMR (Supplementary Fig. 5a), proposing a feedback mechanism between DNAm and gene expression regulation for a minority of the 513 mQTLs (1.3%) and 292 eQTLs (2.3%). A larger number of significant M2E_SMR associations compared to E2M_SMR suggests that the genetic effect of DNAm on gene expression is a more predominant mechanism than the genetic effect of gene expression on DNAm.

We further compared the genes identified in both E2M_SMR and M2E_SMR analysis and discerned 1447 common genes that were common out of 1983 E2M_SMR and 2101 M2E_SMR genes. Notably, ten common genes were enriched (FDR < 0.05) in the glutathione metabolism pathway (Fig. 3e and Supplementary Fig. 5b). The E2M_SMR and M2E_SMR shared genes were also enriched in immune related processes (such as antigen processing and presentation and autoimmune diseases) and metabolic processes (such as glycolysis-gluconeogenesis and metabolism of xenobiotics by cytochrome p450) (Fig. 3e and Supplementary Data 15). For example, for glutathione S-transferase mu 1, *GSTM1*, we detected an association in E2M_SMR with CpG cg24506221 and SNP rs36209093 (SMR $P$ value $< 8.53 \times 10^{-13}$ and HEIDI $P$ value $> 0.05$) and another association in M2E_SMR with SNP rs148490733 and CpG cg24506221 (SMR $P$ value $< 4.60 \times 10^{-11}$ and HEIDI $P$ value $> 0.05$) (Fig. 3f). This causal relationship was further supported by the eQTM analysis, where the CpG cg24506221 located within 200 bp of the TSS of *GSTM1* was negatively correlated with *GSTM1* expression levels (Fig. 3g).

To provide additional support for a shared causal variant between retina mQTLs and eQTLs that co-occur along the genome, we applied Bayesian colocalization, using coloc[43], to all significant eQTL and/or mQTL (EM) variants (Supplementary Fig. 5c). Using a stringent posterior probability of association (PPA) ≥ 0.8, we determined a significant colocalization for 9417 variants and 2423 genes (Supplementary Data 16). Target genes of the colocalizing EM signals were enriched in purine and pyrimidine metabolism pathways (FDR < 0.05) (Supplementary Data 17 and Supplementary Fig. 5d). We further compared EM results from coloc with E2M_SMR and recognized 343 associations in common corresponding to 177 genes (Supplementary Fig. 5e).

## SMR and colocalization of retina mQTLs and eQTLs prioritize causal genes for AMD loci

To test for causal or pleiotropic (hereafter referred to as pleiotropic) relationships between DNAm and AMD risk, we applied SMR to all significant independent retina mQTLs and AMD GWAS summary statistics[26] (Fig. 4a). We identified 28 significant associations between mQTLs and AMD at 7 GWAS loci (SMR $P$ value $< 5.68 \times 10^{-7}$ after Bonferroni correction) (Supplementary Data 18); of which, 5 associations at 3 loci (*KMT2E/SRPK2*, *PILRB/PILRA* and *ARMS2/HTRA1*) passed the

linkage test (HEIDI $P$ value $> 0.05$) and are thus likely to be pleiotropic (Fig. 4b). Similarly, we used our retina eQTLs to test for pleiotropic associations between gene expression and AMD GWAS and identified 14 associations at 7 loci (SMR $P$ value $< 6.41 \times 10^{-6}$ after Bonferroni correction) (Supplementary Data 19); of which, 8 associations at 5 loci (*CFI, C2/CFB/SKIV2L, PILRB/PILRA, RDH5/CD63* and *TMEM97/VTN*) passed the linkage test (HEIDI $P$ value $> 0.05$) (Fig. 4c). In our previously published eQTL study, we identified the same target genes for 4 of the loci, including *CFI, PILRB/PILRA, RDH5/CD63 and TMEM97/VTN*, based on colocalization analysis[32] (Supplementary Data 19). The retina mQTLs and eQTLs proposed the same target genes for one locus *PILRB/PILRA*, and new causal mechanisms and genes for AMD at 3 (*KMT2E/SRPK2, PILRB/PILRA* and *ARMS2/HTRA1)* and 2 (*C2/CFB/SKIV2L* and *PILRB/PILRA*) loci, respectively (see below). For example, at the *PILRB/PILRA* locus, we identified SNP rs11766752 and CpG cg07160278 influencing both DNAm and AMD (SMR $P$ value $= 2.88 \times 10^{-7}$ and HEIDI $P$ value $= 0.09$). cg07160278 is localized in the TSS region of *MEPCE* and *ZCWPW1* genes, and for this CpG we identified eQTM with *PILRA* gene (Fig. 4d). With eQTL and AMD GWAS SMR analysis, we identified SNP rs45451301 at the *C2-CFB-SKIV2L* locus influencing both *DXO* expression and AMD risk (SMR $P$ value $= 2.02 \times 10^{-7}$ and HEIDI $P$ value $= 0.38$) (Fig. 4e). At the *CFI* locus, SMR detected SNP rs7439493 influencing *PLA2G12A* expression and AMD risk (SMR $P$ value $= 2.6 \times 10^{-8}$ and HEIDI $P$ value $= 0.22$) (Supplementary Fig. 6a), as identified in our previous eQTL study[32].

We next used the Bayesian colocalization method eCAVIAR[42] to test whether retina mQTLs and eQTLs share one or more causal variants with known AMD GWAS loci. Significant colocalization would prioritize potential causal mQTLs/eQTLs and target CpGs or genes for AMD. Of 52 independent AMD GWAS variants at 34 loci, we identified significant colocalization (Colocalization posterior probability, CLPP > 0.01) of one or more mQTLs and/or eQTLs with 17 AMD GWAS variants in 10 loci (29.4% of loci) (Supplementary Fig. 6b). An average of $1.87 \pm 0.52$ causal genes per locus were proposed based on mQTLs, $1.4 \pm 0.24$ based on eQTLs, and $2.1 \pm 0.50$ causal genes based on mQTLs and/or eQTLs (Supplementary Data 20). A single causal gene was proposed for 5 (14.7%) of the loci (Supplementary Data 20). Fifteen AMD GWAS variants in 8 (23.5%) loci colocalized with at least one retina mQTLs, and 6 AMD GWAS variants in 5 loci (14.7% of loci) colocalized with at least one retina eQTL (Supplementary Data 20). Only mQTLs colocalized with an AMD signal at 5 loci, with *CFH* as a target gene at the *CFH* locus (Supplementary Fig. 6c, 6d), a lincRNA *LINC01004* at the *KMT2E/SRPK2* locus (Fig. 4f, 4g), *ARHGAP21* and *RNA5SP305* at the *ARHGAP2* locus, *RAD51B* at the *RAD51B* locus, and *MARK4* at the *APOE(EXOC3L2/MARK4)* locus. Two loci had only eQTLs colocalizing with the AMD signal, with *SARM1* and *TMEM199* target genes at the *TMEM97/VTN* locus and *TMEM259* at the *CNN2* locus. At the AMD locus rs147859257/rs2230199 on chromosome 19, both an mQTL (cg12024887 and cg07567260 that mapped to *GRP108*, *MIR6791* and *TRIP10*) and an eQTL (acting on *GRP108*) colocalized with the AMD association signal, suggesting that the causal effect of the colocalizing mQTL in this AMD locus may be acting via altered *GRP108* expression levels (Supplementary Fig. 6e, f, g). Eighteen of the colocalizing genes have not yet been identified as targets at corresponding AMD loci. The mQTL of CpG cg11712338 mapped to *LINC01004* and colocalized with AMD at the *KMT2E/SRPK2* locus (Fig. 4f, 4g). SMR analysis further validated this mQTL as a high confidence causal effect on AMD in this locus (LD $r^2 = 0.98$ between significant mVariant based on SMR and eCAVIAR, rs3214376 and rs6950894, respectively).

## Colocalization analysis of retina mQTLs, eQTLs and GWAS identifies additional AMD genes

To determine shared causal variants between retina mQTLs and eQTLs, and between m/eQTLs and AMD loci and identify novel AMD genes, we applied the colocalization method coloc[43] to the following pairwise

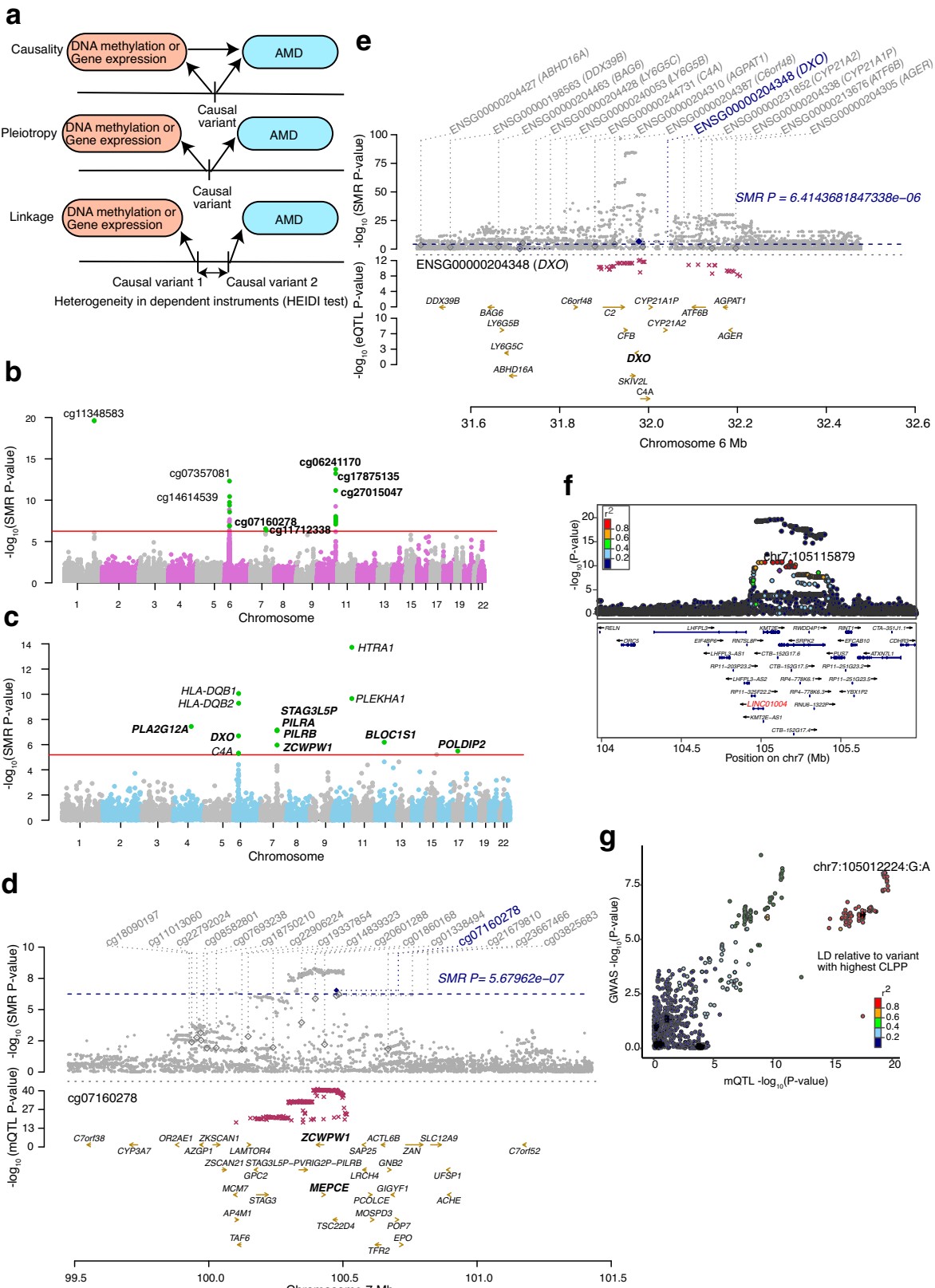

comparisons: AMD GWAS and mQTL (GM), AMD GWAS and eQTL (GE), and mQTL and eQTL (EM) (described above), considering all significant mQTLs/eQTLs genome-wide (Fig. 5a and Methods). We discovered significant GM colocalization results for 69 mVariants and 81 CpG sites corresponding to 58 genes (PPA ≥ 0.8) at 11 AMD loci (Supplementary Fig. 7a and Supplementary Data 21) and significant GE

colocalization for 44 eVariants (Methods) and 32 genes at 3 AMD loci (Supplementary Fig. 7a and Supplementary Data 22). Three colocalizing variants at 3 AMD loci were significant with both GM and GE (Supplementary Data 23). The genes identified in GM and GE were enriched (FDR ≤ 0.05) in immune-related processes, such as antigen processing and presentation (Supplementary Fig. 7b). The GM and GE

**Fig. 4 | Associations between retina DNA methylation, gene expression and AMD GWAS through genotypes. a** Heterogeneity in dependent instruments (HEIDI) model to distinguish pleiotropy from linkage for an observed association between AMD, DNAm and/or gene expression through genotypes. **b** Manhattan plot of SMR tests for association between retina mQTL and AMD GWAS. Shown on each y axis are the $-\log_{10}$(SMR P values) from SMR tests. The red horizontal lines represent the genome-wide significance level (SMR P values = $5.67 \times 10^{-7}$). **c** Manhattan plot of SMR tests for association between retina eQTL and AMD GWAS. The red horizontal lines represent the genome-wide significance level (SMR P values = $5.4 \times 10^{-6}$). **d** Results of variants and SMR associations across retina mQTL and AMD GWAS. The top plot shows $-\log_{10}$(SMR P values) of SNPs comparing mQTL and AMD GWAS. The blue diamonds represent $-\log_{10}$(SMR P value) from SMR tests for associations of mQTL and AMD GWAS with SMR P value = $5.67 \times 10^{-7}$.

The second plot shows $-\log_{10}$(SMR P value) of the SNP associations for DNAm probe cg07160278 from the mQTL data. **e** Results of variants and SMR associations across retina eQTL and AMD GWAS. The top plot shows $-\log_{10}$(SMR P value) of SNPs from eQTL and AMD GWAS. The blue diamonds represent $-\log_{10}$(SMR P value) from SMR tests for associations of eQTL and AMD GWAS with SMR P value = $6.41 \times 10^{-6}$. The second plot shows $-\log_{10}$(SMR P value) of the SNP associations for *DXO* gene from the eQTL data. **f** LocusZoom plot of the retina mQTL associations for cg11712338 (*LINCO1004*). $-\log_{10}$(P value) of retina mQTL with points color coded based on LD ($r^2$) relative to the lead AMD GWAS variant (chr7:105115879:C:T; rs1142) in the locus *KMT2E/SRPK2*. **g** LocusCompare plot comparing $-\log_{10}$(P value) of AMD GWAS to $-\log_{10}$(P value) of retina mQTLs acting on cg11712338 (*LINCO1004*).

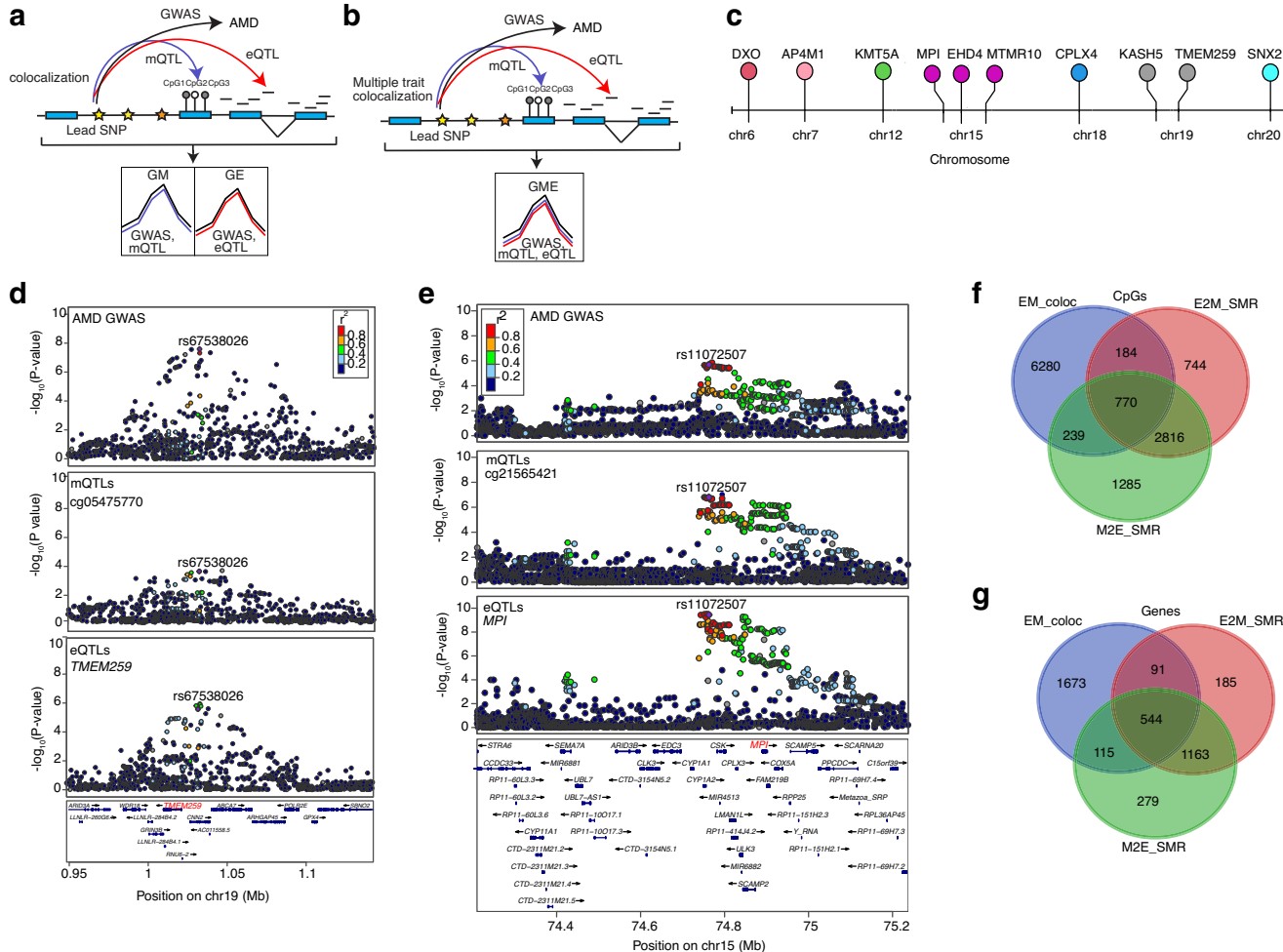

**Fig. 5 | Colocalization analysis among AMD GWAS and retina mQTL and eQTL using Coloc and Moloc. a** Schematic representation of colocalization analysis of AMD GWAS with retina mQTL or eQTL. **b** Schematic representation of multiple trait colocalization analysis of AMD GWAS, retina mQTL and eQTL. **c** Lolliplot depicting the target genes identified in the multiple trait colocalization analysis (moloc) of AMD GWAS, retina mQTL and eQTL. The colors correspond to genes on different chromosomes. **d** LocusZoom plots of AMD GWAS (genotype and GWAS association), CpG cg05475770 mQTLs (genotype and cg05475770 methylation association), and *TMEM259* eQTLs (genotype and *TMEM259* expression association). The y axis shows $-\log_{10}$(P-value) of association tests from GWAS, mQTLs and eQTLs. Points are color coded based on LD ($r^2$) relative to the variant with highest colocalization posterior probability in the locus, rs67538026, identified in moloc analysis of AMD GWAS, mQTLs and eQTLs (GEM). **e** LocusZoom plots of AMD GWAS (genotype and GWAS association), CpG cg21565421 mQTLs (genotype and

cg21565421 methylation association), and *MPI* eQTLs (genotype and *MPI* expression association). The y axis shows $-\log_{10}$(P value) of association tests from GWAS, mQTLs and eQTLs. Points are color coded based on LD ($r^2$) relative to the variant with highest colocalization posterior probability in the locus, rs11072507, identified in moloc analysis of AMD GWAS, mQTLs and eQTLs (GEM). **f** Venn-diagram representing common and unique CpGs identified in associations between retina eQTLs and mQTLs using moloc (EM_coloc), CpGs identified in associations between retina eQTLs and mQTLs using SMR (E2M_SMR), and CpGs identified in associations between retina mQTLs and eQTLs using SMR (M2E_SMR). **g** Venn-diagram representing common and unique target genes identified in associations between retina eQTLs and mQTLs using coloc (EM_coloc), genes identified in associations between retina eQTLs and mQTLs using SMR (E2M_SMR), and genes identified in associations between retina mQTLs and eQTLs using SMR (M2E_SMR).

significant colocalizations detected 42 and 47 new AMD associations, respectively. Considering these new AMD associations, we identified 36 mVariants and 43 CpG sites corresponding to 30 genes in GM and 29 eVariants and 24 genes in GE (Supplementary Data 21 and 22).

To further identify variants associated with DNAm, gene expression and AMD, we applied the multi-trait colocalization method, moloc[44], across AMD GWAS, retina eQTL and mQTL (GEM) signals (Fig. 5b) (see Methods). We obtained strong evidence (PPA ≥ 0.8) at 18 CpG sites and 10 genes (Fig. 5c) where variations in DNAm, gene expression and AMD GWAS were attributed to the same variant (Table 1, Supplementary Data 24). Seven of the significant colocalizing variants were present at 3 known AMD loci (*C2/CFB/SKIV2L*, *PILRB/PILRA* and *CNN2*) corresponding to 3 genes (Fig. 5c), and 11 of the colocalizing variants proposed 7 new AMD associations (Table 2). SNP rs67538026 associated with methylation of CpG cg05475770, *TMEM259* expression and AMD risk was identified at the *CNN2* locus (Fig. 5d), and rs114820183 associated with cg22670819 methylation, *DXO* expression, and AMD risk at the *C2-CFB-SKIV2L* locus. *DXO* was also proposed as a potential AMD gene at this locus by SMR analysis of retina eQTL and AMD GWAS statistics.

Additionally, the moloc analysis identified 7 colocalizing variants that affect DNAm and gene expression of 7 genes at 7 loci that are not reported AMD loci and are therefore suggested as new AMD associations (Supplementary Data 24). These include the SNP rs11072507 associated with methylation of CpG cg21565421, *MPI* expression and AMD (Fig. 5e), and the SNP rs6493454 associated with CpG cg19881011 methylation, *MTMR10* expression and AMD (Supplementary Fig. 7c). We compared CpGs and genes identified in the GM or EM coloc and GEM moloc analyses, and only detected an overlap of 6 CpGs in GM and GME, 4 genes (*KASH5, EHD4, CPLX4 and SNX21*) common in GEM and EM, and 5 genes (*C8orf58, PDZD7, FN3KRP, SLC16A3, SCAF1*) common in GE and EM (Supplementary Fig. 7d, e). We further overlapped CpGs and genes identified in the SMR analysis (E2M_SMR, M2E_SMR) and EM coloc analysis, and noted 770 CpGs and 544 genes in common in all 3 analyses (Fig. 5f, g) (Supplementary Data 25). The CpGs were mapped to potential target genes (mGenes; see Methods), which showed enrichment (FDR < 0.05) in pyruvate metabolism and olfactory signaling pathway (Supplementary Fig. 7f). The results from all colocalization and SMR analyses are summarized in Table 2.

## Integrative analysis with Hi-C retina map improves target gene prioritization

Finally, to prioritize a high confidence list of variants that contribute to DNAm and gene expression variation in retina, we integrated loops, CREs and SEs from our recently published retina Hi-C data[36] with mQTLs, eQTLs, and eQTMs, and their target genes, and variant associations identified in the SMR, eCAVIAR, coloc and moloc analyses. We observed that 73% of mQTLs, 78% of eQTLs and 82% of eQTMs are localized to the A compartment (representing active transcription chromatin regions) compared to 23% of mQTLs, 18% of eQTLs and 16% of eQTMs in the B compartment (representing closed chromatin regions with inactive genes); 4% of e/mQTLs or eQTMs fell in uncategorized chromatin regions (Fig. 6a and Supplementary Data 26). Most of the retinal mQTLs and eQTLs are present within the A compartment providing a direct mechanism to explain the impact of a variant on the mGene or eGene. We next examined the overlap of Hi-C loop foot locations with mQTL and eQTL variants and their target genes, distinguishing promoter mQTLs/eQTLs (±2.5 kb from TSS of target gene) from distal mQTLs/eQTLs (>2.5 kb from TSS) (see Methods), as previously described[53]. We identified 5479 mQTLs overlapping a loop foot; of these, 104 (1.8%) interacted with their mGene promoter (promoter mQTLs), 705 (12.8%) were distal mGene mQTLs, 56 (1%) were promoter non-mGene mQTLs, and 4614 (84.2%) were distal non-mGene mQTLs (Fig. 6b left panel and Supplementary Data 27). Among the 705 distal mGene mQTLs, 2 mQTLs are associated with known AMD loci (*SYN3/TIMP3* and *TRPM3*). For eQTLs, 1721 overlapped a loop foot; of these, 17 (0.9%) interacted with their eGene promoter (promoter eQTLs), 186 (10.8%) were distal eGene eQTLs, 15 (0.8%) were promoter non-eGene eQTLs, and 1503 (87.3%) were distal non-eGene eQTLs (Fig. 6b middle panel and Supplementary Data 28). We similarly tested for overlap of promoter eQTMs (CpGs within ±2.5 kb from target gene TSS) and distal eQTMs (>2.5 kb from TSS) with retina Hi-C

## Table 1 | Significant moloc colocalization results for AMD GWAS, retina eQTL and mQTL

| CpG | CpG genic location | Distance between CpG and its corresponding target gene TSS (bp) | Target gene[a] | chr | SNP | Variant position (hg38) | GWAS locus[b] |
|---|---|---|---|---|---|---|---|
| cg22670819 | 1stExon | −18964 | *DXO* | chr6 | rs114820183 | 31933131 | *C2/CFB/SKIV2L* |
| cg00854166 | Body | 619452 | *AP4M1* | chr7 | rs34130487 | 100161582 | *PILRB/PILRA* |
| cg14736458 | - | −103608 | *AP4M1* | chr7 | rs34130487 | 100161582 | *PILRB/PILRA* |
| cg04039547 | TSS200 | 47175 | *AP4M1* | chr7 | rs34130487 | 100161582 | *PILRB/PILRA* |
| cg00472528 | - | 932958 | *KMT5A* | chr12 | rs34477554 | 123405486 | - |
| cg19881011 | TSS1500 | −40151 | *MTMR10* | chr15 | rs6493454 | 31101742 | - |
| cg21565421 | Body | −589718 | *MPI* | chr15 | rs11072507 | 74762525 | - |
| cg11955166 | - | −1029954 | *EHD4* | chr15 | rs72735670 | 40922983 | - |
| cg13906792 | TSS1500 | 17427 | *MPI* | chr15 | rs11072507 | 74762525 | - |
| cg16646645 | TSS1500 | −39953 | *MTMR10* | chr15 | rs6493454 | 31101742 | - |
| cg14323928 | 5'UTR | −41329 | *MTMR10* | chr15 | rs6493454 | 31101742 | - |
| cg21851553 | TSS200 | −41153 | *MTMR10* | chr15 | rs6493454 | 31101742 | - |
| cg23999607 | Body | 299453 | *CPLX4* | chr18 | rs4940875 | 59463337 | - |
| cg03536881 | TSS200 | −394519 | *KASH5* | chr19 | rs2098709 | 48684412 | - |
| cg05475770 | Body | −441931 | *TMEM259* | chr19 | rs67538026 | 1031439 | *CNN2* |
| cg00752531 | Body | −442683 | *TMEM259* | chr19 | rs67538026 | 1031439 | *CNN2* |
| cg09935308 | Body | −679262 | *TMEM259* | chr19 | rs113772652 | 1031551 | *CNN2* |
| cg03641251 | Body | −363452 | *SNX21* | chr20 | rs4810499 | 46340784 | - |

[a]Target gene inferred from colocalizing eQTL.
[b]Nearest gene/s to lead GWAS variant. TSS, transcription start site.

**Table 2 | Summary of colocalization and SMR analyses of retina eQTL, mQTL and AMD associations**

| Method | mQTL -> eQTL | eQTL -> mQTL | eQTL -> AMD GWAS | mQTL -> AMD GWAS | mQTL <-> eQTL <-> AMD GWAS |
|---|---|---|---|---|---|
| SMR[a] | 6592 associations (5175 mQTLs, 5012 eQTLs) and 2101 genes (Common with eQTL -> mQTL: 554 genes) | 5805 associations (2256 eQTLs, 5612 mQTLs) and 1983 genes (Common with mQTL->eQTL: 554 associations,1447 genes) | 8 associations, 5 GWAS loci (CFI, C2/CFB/SKIV2L, PILRB/PILRA, RDH5/CD63, TMEM97/VTN) | 5 associations, 3 GWAS loci (KMT2E/SRPK2, PILRB/PILRA, ARMS2/HTRA1) | - |
| coloc | 5175 eQTLs and 2423 genes (343 associations and 177 genes common with SMR eQTL -> mQTL) | 9417 variants and 2423 genes (343 associations and 177 genes common with SMR eQTL -> mQTL) | 44 eVariants and 32 genes at 3 AMD loci; 47 new AMD associations (29 eVariants and 24 genes) | 69 mVariants, 81 CpG sites corresponding to 58 genes, at 11 AMD loci; 42 new AMD associations (36 mVariants and 43 CpG sites corresponding to 30 genes) | 3 colocalizing variants, but different target genes overlapped between eQTL -> AMD GWAS and mQTL -> AMD GWAS, at 3 AMD loci (PILRB/PILRA, TMEM97/VTN, and CNN2). |
| eCAVIAR | - | - | 5 loci (2 loci colocalizing only with eQTL: SARM1 and TMEM199 target genes at TMEM97/VTN locus and TMEM259 at CNN2 locus) | 8 loci (5 loci colocalizing only with mQTL: CFH as target gene at CFH locus, LINC01004 at KMT2E/SRPK2 locus, ARHGAP21 and RNA5SP305 at ARHGAP2 locus, RAD51B at RAD51B locus, and MARK4 at APOE(EXOC3L2/MARK4) locus) | - |
| moloc | - | - | - | - | 18 colocalizing eQTL and mQTL variants for 18 CpG sites and 10 genes in 3 known AMD loci (DXO in locus C2/CFB/SKIV2L, AP4M1 in locus PILRB/PILRA and TMEM259 in locus CNN2) and 7 new AMD associations |

[a]SMR results that pass HEIDI linkage test $P > 0.05$.

loop foot locations. We identified 3395 eQTMs overlapping a loop foot; of these, 24 (0.7%) interacted with their target gene promoter (promoter-eQTMs), 282 (8.3%) were promoter non-target gene eQTMs, 195 (5.7%) were distal target gene eQTMs, and 2894 (85.2%) were distal non-target gene eQTMs (Fig. 6b right panel and Supplementary Data 29).

We next evaluated the overlap of retina mQTLs, eQTLs and eQTMs with retinal CREs and SEs that were identified based on epigenetic marks[36] (see Methods). We discovered 6834 CRE-mQTLs and 2610 SE-mQTLs; of which, 2048 (29.9%) and 271 (10.3%) were promoter-mGene mQTLs, respectively (Supplementary Fig. 8a left panel and Supplementary Data 30). For the retina eQTLs, we ascertained 1953 CRE-eQTLs and 773 SE-eQTLs overlaps; of which, 609 (31.1%) CRE-eQTLs and 84 (10.8%) SE-eQTLs were promoter-eGene eQTLs (Supplementary Fig. 8a middle panel and Supplementary Data 31). For eQTMs, we uncovered 8996 CRE-eQTMs and 1392 SE-eQTMs; of which, 668 (7.4%) CRE-eQTMs and 88 (6.3%) SE-eQTMs were promoter-eQTMs (Supplementary Fig. 8a right panel, and Supplementary Data 32).

We further integrated Hi-C loops, CREs and SEs with target genes, and variant associations identified in our SMR, eCAVIAR, coloc and moloc analyses. In SMR analysis of mQTL and AMD GWAS with CRE, SE and loops, we identified target genes for 3 associations (MEPCE; ZCWPW1 and HTRA1) at 2 AMD loci (PILRB/PILRA, ARMS2/HTRA1) (Supplementary Data 33). For SMR analysis of eQTL and AMD GWAS with CRE, we detected target genes for 2 associations (STAG3L5P and BLOC1S1) at 2 AMD loci (PILRB/PILRA and RDH5/CD63) (Supplementary Data 34). In eCAVIAR analysis of mQTL and AMD GWAS with CRE and loops, we found target genes (ARHGAP21, KMT2E and LINC01004) for 2 colocalizations at 2 loci (ARHGAP21 and KMT2E/SRPK2) (Supplementary Data 35). For E2M_SMR associations, we identified targets genes for 1058 associations with CRE, 538 associations with SE, and 186 associations with loops (Supplementary Data 36). These analyses also identified high confidence causal links with Hi-C loops between retina eQTLs and mQTLs, and 1 eQTL/mQTL association with ALDH2 identified in E2M_SMR (Fig. 6c). Similarly, for M2E_SMR associations, we identified targets genes for 609 associations with CRE, 516 associations with SE, and 187 associations with loops (Supplementary Data 37). We determined 3 mQTL/eQTL associations with GSTP1 identified in M2E_SMR (Figs. 6d), and 4 eQTL/mQTL and mQTL/eQTL associations with EML1 noted in E2M_SMR and M2E_SMR (Supplementary Fig. 8b). For coloc EM colocalizations, we identified target genes for 790 colocalizations with CRE, 450 colocalizations with SE and 216 colocalizations with loops (Supplementary Data 38). We also discovered a target gene for coloc EM colocalization in LPIN1 gene with Hi-C loops (Supplementary Fig. 8c).

### Genes and pathways influenced by DNA methylation and associated with AMD

To obtain a non-redundant set of genes, we merged the target genes identified in mQTL and AMD GWAS analyses using SMR, eCAVIAR, coloc and moloc (Table 2). We identified 4 target genes at 3 loci by SMR, 15 target genes at 8 loci using eCAVIAR, 58 target genes at 11 loci using coloc, and 10 target genes at 3 loci using moloc. By combining the target genes from the four methods, we identified a total of 87 non-redundant genes that are influenced by DNAm (Supplementary Data 39). Fifty of these genes fall in the reported AMD GWAS loci and 37 are target genes in potentially new AMD associations. These genes belong to a range of biological processes including mTOR signaling and RHOF GTPase cycle (Reactome, gProfiler adj. $P < 0.05$), and regulation of actin cytoskeleton reorganization and glycosylation (Gene Ontology, GeneEnrich adj. $P < 0.05$).

## Discussion

DNA methylation (DNAm) has regulatory roles during development, aging and disease[11]. The relationship between DNAm and gene

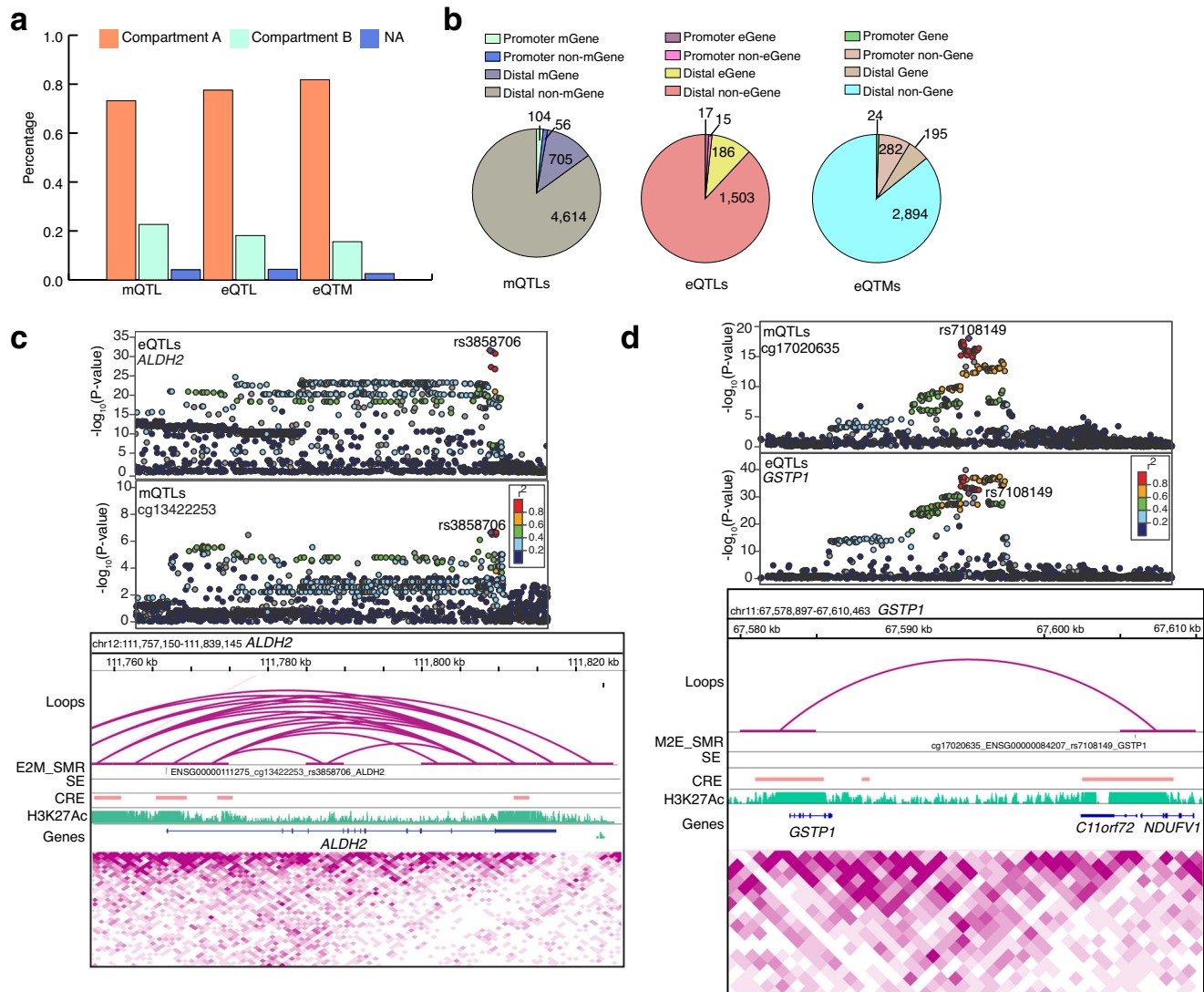

**Fig. 6 | Hi-C retina data enables target gene and variant prioritization. a** Proportion of unique retina mQTLs, eQTLs and eQTMs overlapping each chromatin compartment. **b** Number of unique retina mQTLs, eQTLs and eQTMs overlapping with a loop that are in contact with variants located within ±2.5 kb of the mGene/eGene/Gene TSS were identified as promoter mQTLs/eQTLs/eQTMs, while those located >2.5 kb from the mGene/eGene/Gene were identified as distal mQTLs/eQTLs/eQTMs. **c** Upper panel: LocusZoom plots of *ALDH2* eQTLs (genotype and *ALDH2* expression associations) and CpG cg13422253 mQTLs (genotype and cg13422253 methylation associations). The y axis shows −log₁₀(P value) of eQTL and mQTL association tests. Points are color coded based on LD (r²) relative to the highlighted variant rs3858706, the most significant variant in E2M_SMR analysis of eQTLs and mQTLs. The lower panel includes associations identified in

E2M_SMR for *ALDH2* gene. Tracks represent the retina chromatin loops, E2M_SMR significant variant, SEs, CREs, H3K27Ac coverage, genes, and Hi-C physical contact maps identifying TADs. **d** Upper panel: LocusZoom plots of CpG cg17020635 mQTLs (genotype and cg17020635 methylation associations) and *GSTP1* eQTLs (genotype and *GSTP1* expression associations). The y axis shows −log₁₀(P value) of association tests from mQTLs and eQTLs. Points are color coded based on LD (r²) relative to the highlighted variant rs7108149, the most significant variant in M2E_SMR analysis of eQTLs and mQTLs. The lower panel includes associations identified in M2E_SMR for *GSTP1* gene. Tracks represent the retina chromatin loops, M2E_SMR variant, SEs, CREs, H3K27Ac coverage, genes, and Hi-C physical contact maps. CRE *Cis*-regulatory element, SE Super-enhancer, TAD Topologically associating domain.

expression is complex and influences diverse biological pathways. DNAm can be mitotically inherited and can act as an intermediary molecular link between genetic variation and tissue-specific transcriptional regulation, thereby providing a mechanism for long-term maintenance of cellular identity[10]. As most of the GWAS signals map to non-coding regions, their functional significance may be best explained by regulatory effects on common human traits and diseases. Non-coding genetic variation also contributes to the establishment of DNAm patterns, independently or together with environmental exposures[54]. In the present study, we have performed an integrative genetic, transcriptomic, and epigenetic analysis identifying genes and pathways that are regulated in human retina under normal conditions and/or may be influenced by advancing age and environmental factors.

We support this proposal by identifying mQTLs that may be causal to eQTLs and vice versa using Mendelian randomization and colocalization analyses. The strength of our study is that we applied two different complementary approaches[55] to testing whether mQTLs, eQTLs and/or AMD associations share a common causal variant and whether their association is causal or pleiotropic as opposed to being confounded by linkage of distinct causal variants. The 87 genes and corresponding pathways, we identified here, are modulated by DNAm and may be suggestive of disease progression and pathology.

DNAm can have a dual interplay by being a cause and a consequence of gene expression changes[56]. We discovered genetic associations with DNAm for over 36,000 CpG sites that mapped to about 72.6% of retina-expressed genes. Notably, approximately two-third of

these associations are likely unique to the retina and not observed in brain, muscle, blood, or adipose tissue. We also identified genetic associations with gene expression for 9395 (54%) of retina genes. Retina mQTLs were primarily detected in open chromatin regions, transcription factor (TF) binding sites, enhancers, and synonymous variants. In contrast, the retina eQTLs showed the highest enrichment in splicing regions, 5' and 3' UTR, TF binding sites, and other regulatory regions. Interestingly, the retina mQTLs and eQTLs tended to affect genes in divergent biological processes.

Using multiple algorithms, our maps of mQTLs and eQTLs integrated with AMD GWAS have provided detailed information about genotype changes that mediate DNAm and gene expression in the retina under non-disease conditions and in relation to AMD. Even though mQTLs identified here are limited by sample size compared to the published eQTL studies[32–34], we have reported more target mGenes than eGenes associated or colocalized with AMD loci, indicating a larger impact of epigenetic changes due to environmental factors and advanced age. Our work suggests that mQTLs uncover novel AMD-associated molecular mechanisms that have been missed by eQTLs. In addition, integration with retina Hi-C data has yielded supportive evidence of regulatory mechanisms and helped in determining high confidence target genes for mQTLs and eQTLs. For example, we ascertained 5 mQTLs and 1 eQTL affecting *PARK7*, which is involved in synaptic signaling and protecting neurons against oxidative stress and cell death[57]. We also recognized a CRE overlapping these 5 mQTLs using retina epigenetic marks. A previous study of retina from *PARK7/DJ-1* deficient 3- and 6-month-old mice showed that knock-out of *PARK7* leads to changes in electroretinogram (ERGs), which measures the electrical activity of retina in response to light stimulus and increased oxidative stress, suggesting a possible connection to aging and AMD[58].

We characterized eQTMs for over 10,000 CpG sites, which account for 1.4% of all methylated CpGs, yet map to 75% of genes expressed in the retina. Our eQTM analysis revealed that the majority of CpGs affecting gene expression are localized in the TSS region, as observed in other tissues[20,21,37]. Though the methylated CpGs in eQTMs are enriched in TSS regions, 10-fold more eQTMs targeted a distal gene rather than a proximal gene supported by Hi-C chromatin loops; these results suggest that many CpGs act as enhancers regulating the associated gene. Interestingly, CpG methylation has been commonly associated with downregulation of gene expression; nonetheless, almost 45% of eQTMs we identified are positively associated with gene expression, consistent with previous eQTM studies[20,21,37]. We also note that some TFs prefer to bind to methylated CpGs, and methylation does not always equal repression[59]. Previous studies had not considered the effect of chromosome size on eQTMs. Notably chromosomes 16, 17, 19 and, 22, though relatively smaller in size, are gene rich[60]. We thus inspected the eQTM distribution by chromosome and observed that few of these chromosomes possessed a greater number of eQTMs. For example, CpGs that regulate more than 10 genes clustered on the p arm of chromosome 16. A few of these genes contribute to telomere maintenance. A larger region on chromosome 16p has been associated with autism and decreased expression of genes enriched in the telomeric region[61]. Examination of retina Hi-C data showed that 82% of the eQTMs are localized to the A compartment, i.e., the active compartment. Retinal eQTMs exhibited enrichment in energy metabolism, transcription, and protein localization to the endoplasmic reticulum, highlighting both the susceptibility of these pathways to environmental influence through methylation as well as their adaptive nature. Thus, eQTM studies complement findings from mQTLs and eQTLs and provide novel insights into retina homeostasis and disease.

Genetic variations in Glutathione-S-Transferases (GSTM) have been associated with cancer[62], atherosclerosis[63], and early onset of Parkinson's disease[64]. DNAm-mediated regulation of *GSTM* genes is observed in divergent pathologies including cancer[65], endometriosis[66], and aging of the lens[67]. The retina has a well-established glutathione antioxidant system, and the enzymes involved in glutathione metabolism protect retina from continuous exposure to reactive oxygen species, detoxification of many endogenous compounds and xenobiotics, and reversible modification of proteins protect them during oxidative stress[68]. Previous studies have reported upregulation of glutathione proteins in AMD retinas[69] and significantly greater glutathione depletion in retinal pigment epithelium (RPE) cells from AMD donors[70]. Notably, aging rod photoreceptors also display hypermethylation of *GSTM*2, *GSTM*5 and *GSTM*6 promoters[38]. Based on Mendelian randomization, ten putative target genes of mQTLs are causal to or share an association with eQTLs and vice versa and are enriched in the glutathione metabolism pathway. Thus, genetic and environmental control of DNAm levels at *GSTM* genes could impact retinal physiology during aging and disease.

Previous studies have assessed the sharing of a causal variant between expression and methylation QTLs in various tissues using colocalization analysis that assumes a single causal variant (coloc) and/or summary statistic-based Mendelian randomization (SMR)[20,21,37]. Given the widespread allelic heterogeneity observed in e/mQTLs[5,21,42], in addition to using coloc, we applied colocalization methods that account for multiple causal variants (allelic heterogeneity) in m/eQTL and GWAS loci (eCAVIAR and moloc), as well as SMR that tests whether co-occurring mQTL, eQTL and AMD associations are causal or pleiotropic vs. induced by linkage. Given the differences in the underlying assumptions of the different colocalization and MR methods[55], applying a range of methods has enabled us to cast a wide net and prioritize a comprehensive list of candidate mQTL/eQTL and AMD genes for follow-up functional studies, and to detect new AMD associations that may be mediated by eQTLs or mQTLs. As every method is unique and has somewhat different underlying assumptions, we could demonstrate commonality only for a few results, which we present as higher confidence causal relationships. For example, we detected one common mQTL of CpG cg11712338 at the *KMT2E/SRPK2* locus between SMR associations and eCAVIAR colocalization results for mQTL and AMD GWAS. For pairs that are not common between colocalization and SMR analyses, hidden confounders may not be well captured by the covariates we adjusted for, or this could be due to the limited power of each method. Clearly, since each of these methods have their limitations, experimental testing will be needed to confirm causality.

Newer chromatin interaction methods, developed in the last decade, have implicated long-range looping interactions between regulatory elements and promoters[71]. Integration of gene expression and AMD GWAS with a retina chromatin map was recently reported by our group[36]. Incorporation of retina Hi-C data into the current study allowed us to identify a high confidence set of target genes that are linked to mQTLs, eQTLs, and eQTMs based on the SMR, eCAVIAR, coloc, and moloc methods. For example, we show a remarkable long-range interaction between the variant rs3858706 and the target gene *ALDH2*, whose association was evident in the E2M_SMR analysis. *ALDH2* is upregulated in AMD retinas, indicating that it responds to changes in cellular redox status[72]. We also identified the target gene *EML1* for the variant rs11625037 with retina loops, SE and CREs, whose associations were found in E2M_SMR and M2E_SMR analysis. *EML1* is a microtubule-associated protein-like gene in which rare mutations are linked to Usher syndrome (USH), a group of genetic disorders manifesting congenital deafness, retinitis pigmentosa, and vestibular dysfunction[73]. We thus establish how integration of mQTLs/eQTLs and eQTMs with Hi-C data can further facilitate the prioritization of candidate genes for disease.

Our study represents the first to integrate epigenome and transcriptome dysregulation in the retina with genetic risk of disease, such as AMD. We have discovered multiple genes and CpG sites that show association and colocalization at a shared genetic variant with known

AMD GWAS loci. Several of the associated target genes we identified have not yet reached genome-wide significance in the largest published AMD GWAS[26], demonstrating the ability of the current m/eQTL study to propose novel genes related to this disease. The addition of a retina chromatin map has provided further insights into genes and biological mechanisms driving the genetic associations. It is important to note that different omics datasets and methods we applied provide both complementary and confirmatory information for target prioritization. We are especially interested in gene expression and DNAm levels measured in the retina since epigenome is a reversible system that can be influenced by diverse environmental factors, such as smoking and diet[13]. We hypothesize that future studies of DNAm changes with age and AMD using a larger sample size could potentially uncover joint contributions of genetic variability, aging, and environment on AMD risk. Finally, incorporating additional QTL types of molecular traits, such as splicing QTL (sQTL), protein QTL (pQTL), chromatin accessibility QTL (caQTL), and histone modification QTL (hQTL) are expected to yield greater understanding of retinal function and disease pathology.

In conclusion, our multi-stage analyses incorporating GWAS, mQTL and eQTL data with multiple colocalization and MR methods provide a statistically comprehensive analytical approach that integrates genomic, transcriptomic and epigenomic information to elucidate regulatory mechanisms for AMD. Using this integrative analysis, we have identified genes in pathways associated with AMD, including glutathione metabolism pathway, mitochondria, immune-related processes, metabolic processes, and mTOR signaling. Our results contribute to a better understanding of the retinal DNA methylome and its relationship with both the transcriptome and AMD. The analyses we perform here can be applied to any other tissue, phenotype, or disease, using the suite of tools that we have used. Altogether, our study provides the research community with a valuable resource of integrative genetic, transcriptome and epigenome dataset in the retina to investigate causal mechanisms of complex traits such as AMD from a multiomic standpoint and can be expanded as more data becomes available in the future.

## Methods

### Datasets
Two DNAm datasets were produced in this study, the Bethesda dataset that consists of 96 retina samples and the Cologne and Munich dataset that consists of 64 retina samples.

### Postmortem retina samples and DNA methylation
For the Bethesda dataset, postmortem peripheral retina samples from 96 deidentified donors were procured by the Minnesota Lions Eye Bank in accordance with the tenets of the Declaration of Helsinki and following informed consent from the donors or their family. Given that deidentified post-mortem samples were used, the study was exempted by the institutional review boards of the University of Minnesota and National Eye Institute, National Institutes of Health. Eyes were enucleated within 4 h of death and stored in a moist chamber at 4 °C until retinal dissection was performed. The post-mortem interval (PMI) for tissue dissection ranged between 5 and 24 h[32] and was corrected for in the eQTL and eQTM mapping through surrogate variable analysis (see below). Retina samples were graded on the Minnesota Grading System (MGS) as previously described[74,75] ranging from 1: no AMD, (n = 40) 3: intermediate AMD (n = 28), and 4: late AMD (either CNV and/or GA) (n = 28) (Supplementary Data 1). Retinal samples were flash frozen in liquid nitrogen and stored at −80 °C until further processing. DNA were isolated from homogenized retina tissue in TRIzol® (Invitrogen, Carlsbad, CA) as per the previously published protocol[76]. DNA was quantified using the QuantiFluor® dsDNA System (Promega, Madison, WI). DNA from each retina sample (500 ng) was used for the MethylationEPIC BeadChip.

For the Cologne and Munich dataset, 64 peripheral retina samples (MGS1 (Controls = 63) and MGS4 (Advanced AMD = 1)) were collected at the Ludwig-Maximilians-University (LMU) Munich and the University of Cologne Eye Bank after informed consent from the donor or next of kin was obtained. This was done in full accordance with the tenets of the Declaration of Helsinki. The tissue and data collections and the subsequent study was approved by the local Ethics Boards at the LMU (Application nr. MUC73416) and the University of Cologne (application nr. 14-247), respectively.

At the LMU, whole globes were prepared as described before[77]. After cleaning of the whole eye in 0.9% NaCl solution, immersing in 5% polyvinylpyrrolidone-iodine, and rinsing again with NaCl solution, the globe was prepared and vitreous was removed. Subsequently, the retina was harvested with two sterile forceps and dissected at the optic nerve head. Thereafter, the retina was transferred to a 1.5 ml Eppendorf cup and stored at −80 °C. Donor eyes from the eye bank of the University of Cologne were processed within 6 h of death and retinal dissection was performed. Retinal samples were then flash frozen in liquid nitrogen and stored at −80 °C until further processing. Genomic DNA was extracted using the salting-out method described elsewhere[78]. Methylation analysis was conducted with the MethylationEPIC BeadChip on 64 retina samples. Processing of the MethylationEPIC BeadChip was performed at Life & Brain GmbH (Venusberg-Campus 1, Bonn; Germany). Details of the 64 retina samples including donor sex, age and MGS grade are described in the Supplementary Data 1.

### Genotyping data processing and quality control for mQTL analysis
Genotyping data for the methylation quantitative trait locus (mQTL) analysis was taken from our two previously published studies, which include the Illumina Infinium CoreExome-24 bead array (Illumina, San Diego, CA) array for the Bethesda dataset (n = 96) and the Axiom™ Precision Medicine Research Array for the Cologne and Munich dataset (n = 64)[32,34]. The independently imputed genotype calls of each dataset to 1000 Genomes Project Phase 3 were merged into a single VCF file, followed by standard quality control (QC) steps, including removing variants with low imputation quality (INFO < 0.3) or that failed Hardy-Weinberg Equilibrium ($P < 1 \times 10^{-6}$) as described[34]. This resulted in 8,899,938 common (minor allele frequency (MAF) > 5%) genetic variants used for mQTL analysis. Principal component analysis (PCA) was applied to the imputed genotype variants after performing linkage disequilibrium (LD) pruning with variants MAF > 5% and high call rate of >98%, using Eigenstrat (v6.1.4) (https://github.com/DReichLab/EIG) to identify outlier samples and principal components (PCs) that captured most of the genotype variation between the samples[79,80] (Supplementary Fig. 1a). Furthermore, kinship analysis was performed with the kinship option of qctool v2 (https://www.well.ox.ac.uk/~gav/qctool_v2/) and related samples (kinship coefficient > 0.8) as well as samples with contradictions in inferred and reported sex according to the Axiom™ analysis suite version 3.1 were excluded. Overall, 4 samples were removed from the Cologne and Munich dataset. After sample QC, a total of 156 samples from both the datasets were considered for mQTL analysis and the top 10 genotype PCs were used in the mQTL model described below (Supplementary Fig. 1b).

### Quality control and normalization of EPIC BeadChip
Raw idat files from both studies were similarly processed using minfi (v1.42.0) pipeline[81] (https://bioconductor.org/packages/release/bioc/html/minfi.html). Clustering of the median methylated and unmethylated signal showed 4 outliers, which were subsequently removed, leaving 152 samples for further analysis. Inspection of the bisulfite conversion plots indicated poor bisulfite conversion to have caused these outliers. We used the following stringent QC criteria for the CpG

sites: i) Probes with a detection p-value below <0.01 in 10% of samples and with a bead count of <3 in 10% of samples were removed, ii) Potentially cross-hybridizing probes[82]; Probes that include SNPs and probes on the sex chromosomes were removed. The data underwent subsequent quantile normalization using the preprocessQuantile functions from the *minfi* R package. Normalized M-values (defined as log2(M/U), where *M* refers to the DNAm probe intensity and *U* refers to the unmethylated probe intensity) were further used for statistical testing and PCA analysis. We performed PCA of the M-values across the 152 samples and found a batch effect between the samples collected at three different sites (Supplementary Fig. 1c). To correct for DNAm variation due to batch effects and avoid confounding with AMD grade, we applied the unsupervised surrogate variable analysis (SVA) method (v3.38.0)[83] (https://bioconductor.org/packages/release/bioc/html/sva.html) to identify surrogate variables (SVs). The SVs are known and hidden artifacts (confounding factors) impacting analysis. We used the following model:

DNA methylation ~ Sample collection site + MGS + sex + age

We identified 38 surrogate variables (SVs), and these were applied to the M-values using the removeBatchEffect function using limma (v3.46.0) (https://bioconductor.org/packages/release/bioc/html/limma.html) in R accounting for known batch effects. We repeated PCA with batch corrected M-values and observed that the samples were clustering together and found no outliers (Supplementary Fig. 1d). Batch corrected M-values were used in mQTL analysis.

### Differential DNA methylation between AMD grades
We tested for differential DNA methylation of 749,159 CpG sites between control (MGS1) and intermediate (MGS3) or advanced (MGS4) AMD retina samples, using limma (v3.46.0) (https://bioconductor.org/packages/release/bioc/html/limma.html), adjusting for covariates: MGS grade, sample collection site, age, sex and 38 DNAm SVs. Only one CpG (cg10546045) passed Benjamini-Hochberg FDR < 0.05 for MGS3 vs. MGS1.

### Methylation quantitative trait loci (mQTL) mapping
The analysis included 152 samples with their corresponding genotype and M-value. QTLtools[45] (v1.3.1) software suite was used for discovery of methylation quantitative trait loci (mQTL) with 8,899,938 genotyped and imputed common variants and normalized M-values for 749,158 CpG sites. Surrogate variable analysis was applied to the M-values across all 152 samples using the sva package[83] (https://bioconductor.org/packages/release/bioc/html/sva.html). To determine the number of SVs to include in the mQTL regression model, we applied an optimization approach used in GTEx[5], identifying the minimum number of SVs that maximize the number of mQTLs. Testing up to 38 SVs in increments of three SVs in mQTL mapping, we observed that 15 SVs maximized the number of CpG sites with a significant mQTL and the number of mQTLs at False Discovery Rate (FDR) ≤ 0.05 (Supplementary Fig. 2a). The top 15 SVs were sufficient to adjust for batch effect of M-values between studies (Supplementary Fig. 1e). Sample collection site, disease status (MGS level), age, sex, population stratification (top 10 genotype PCs), and 15 SVs were used as covariates in the mQTL mapping. Empirical *p* values were calculated for each CpG by using permutation analysis and fitting a beta distribution on the genotype regression coefficient *p* values of all variants tested per CpG, as described in QTLtools[45,46] (https://qtltools.github.io/qtltools/). Storey & Tibshirani FDR procedure, which is a part of the QTLtools software suite, was applied to the empirical mQTL *p* values to correct for the total number of CpGs tested. Significance was determined at FDR ≤ 0.05. Potential target genes of the mQTLs (mGenes) were defined using Illumina's annotation[84] that is based on the UCSC RefGene mapping of CpGs to genes. All the findings are reported based on singleton CpGs.

### Genomic annotation of mQTLs
Mapping and annotation of CpGs on the MethylationEPIC BeadChip has been described previously[84]. Genomic features of all CpGs tested for mQTLs and CpGs with significant mQTLs (FDR ≤ 0.05) were annotated in gene body, first exon, intergenic, 3 prime untranslated regions (3' UTR), 5 prime untranslated regions (5' UTR), 0–200 bases upstream of transcription start sites (TSS200), and 201–1500 bases upstream of transcription start sites (TSS1500). Enrichment of CpGs in these genomic features was assessed by calculating the ratio of the number of CpGs with a significant mQTL in a specific feature divided by the total number of CpGs tested for mQTLs in the same feature.

### mQTL tissue specificity
To gain insight into the tissue specificity of the discovered mQTLs for peripheral retina, we examined mQTLs from adipose (n = 119)[49], blood (n = 614)[50], endometrium (n = 66)[51], frontal cortex (n = 526)[52] and skeletal muscle (n = 282)[20]. For comparative analysis, we obtained the significant mQTLs (variant-CpG pairs) for adipose, blood, endometrium, skeletal muscle and frontal cortex tissues from the corresponding studies. Since for skeletal muscle only the nominal results were provided without significance information, we considered mQTL (variant-CpG pairs) with *P* value < 0.05 as significant in our analysis.

### RNA-seq data
Gene expression data was taken from our previously described published studies for 406 RNA-seq retina samples from the Bethesda study[32] and 56 RNA-seq retina samples from the Cologne and Munich study[34].

### Genotyping data processing and quality control for eQTL analysis
Genotyping and 1000 Genomes Project-imputed data for the expression quantitative trait locus (eQTL) analysis was taken from our previously published studies of 406 samples[32]. Three samples were further removed due to high per-chromosome missingness rate (>10%) and an average missingness rate above 3% across all chromosomes. Genotype PCA was performed on LD-pruned SNPs using Eigenstrat (v6.1.4). A total of 403 samples and 8,924,684 imputed variants post-QC were considered for eQTL analysis and the top 10 genotype PCs were used in the eQTL regression model described below.

### Batch correction for gene expression data
To correct for hidden factors and batch effects and avoid confounding with AMD grade, we applied the supervised SVA method[85] (v3.38.0) (https://bioconductor.org/packages/release/bioc/html/sva.html) to the gene expression values, including known batch effects in the model as follows:

Gene expression ~ Sample collection site + MGS + sex + age

The set of negative-control genes for supervised SVA were taken from our previously published RNA-seq study[32]. We identified 21 significant SVs, and these were further used as covariates in eQTL analysis.

### Expression quantitative trait locus (eQTL) mapping
The analysis included 403 samples with their corresponding genotype (MAF>1%) and retina gene expression data for 17,382 genes expressed at ≥1 CPM in at least 10% of the retina samples. The gene expression values were normalized using Trimmed Mean of M-values (TMM) in Counts per Million (CPM) using edgeR[86,87] (version 3.32.1). QTLtools[45] (https://qtltools.github.io/qtltools/) software suite was used for discovery of expression quantitative trait loci (eQTL). Surrogate variables determined by the sva package (https://bioconductor.org/packages/release/bioc/html/sva.html), disease status (MGS level), age, sex, population stratification (top 10 genotype PCs), were used as covariates. We took the optimization approach to identify the minimum number of SVs that maximize the number of eQTLs and eGenes (target

gene of eQTL). We tested 21 SVs as covariates during the eQTL mapping and 21 SVs maximized the number of eGenes and eQTLs at FDR ≤ 0.05 (Supplementary Fig. 2b). eVariants were defined as variants significantly associated with gene expression changes correcting for multiple testing of variants per gene at FDR ≤ 0.05, as implemented in QTLtools[45,46]. Empirical *P* values were calculated per eGene using beta distribution fit with permutations on the genotype regression coefficient *p* values of all variants tested per gene, as described in QTLtools[45]. Storey & Tibshirani FDR procedure, which is part of the QTLtools software suite, was applied to the empirical p-values of all eGenes to correct for the number of genes tested. Significance was determined at FDR ≤ 0.05.

### Functional enrichment of mQTLs and eQTLs

Enrichment of mQTL and eQTL variants in functional genomic annotations was assessed using TORUS[47,48] (https://github.com/xqwen/torus) and the following command: torus -d $qtl_statistics -annot $annotation_file -est --no_dtss --fastqtl. TORUS was run on all significant variant-CpG and variant-gene pair summary statistics of all significant CpGs and eGenes, respectively, at FDR < 0.05. All mQTLs found in retina, and mQTLs that are specific to retina compared to adipose, blood, endometrium, frontal cortex, and skeletal muscle were tested. The functional annotations analyzed where based on Ensembl's Variant Effect Predictor (VEP) (https://github.com/Ensembl/ensembl-vep) and Loss-Of-Function Transcript Effect Estimator (LOFTEE) (VEP v100, GENCODE v34) annotations of all variants in the imputed genotype array VCF. Distinct consequences from the annotated VCF file were extracted using BCFtools (v1.11) (http://www.htslib.org/download/) and a presence/absence matrix (PAM) was created. Any annotations with less than 100 variants were removed (lower bound requirement of TORUS). The PAM matrix together with reformatted FastQTL[46] (https://github.com/francois-a/fastqtl) files were used as input to TORUS for the QTL functional enrichment analysis. TORUS computed a point estimate of fold-enrichment and 95% confidence intervals (CIs) per annotation and QTL type. Annotations with a lower bound 95% CI value above zero were considered significant. ggforestplot (v0.1.0) (https://github.com/NightingaleHealth/ggforestplot) in R 3.5.0 was used to plot the enrichment results. Due to large CI variations, annotations (consequences) with less than 550 variants were removed from the forest plot for mQTLs and eQTLs. The proportion and number of mQTLs and eQTLs that fell within each functional annotation were computed.

### Expression quantitative trait methylation (eQTM) analysis

We tested for associations between methylation of CpG sites and expression levels of genes whose TSS was within a ± 1 Mb window of each CpG site. eQTM analysis was performed using a linear regression model implemented in R. We analyzed 152 retina samples used in the mQTL analysis with their available gene expression data from RNA-seq[32,34] (96 RNA-seq samples from the Bethesda study and 56 RNA-seq samples the Cologne and Munich study). To correct for batch effects in the RNA-seq data for 152 samples and avoid confounding by AMD grade, surrogate variables (SVs) were identified and estimated for known batch effects and latent factors using supervised SVA method[85] (v3.38.0) (https://bioconductor.org/packages/release/bioc/html/sva.html) and the following model:

Gene expression ~ Sample collection site + MGS + sex + age

A list of negative control genes for the supervised SVA were taken from our previously published RNA-seq study[32]. We identified SVs that were used as covariates in eQTM analysis.

For eQTM analysis, we considered a linear regression model with known covariates, including MGS grade, sex, age and sample collection site, and inferred covariates, where gene expression values were rank-based inverse normalized, and the DNAm values were inverse normalized CpG M-values. The following covariates were included:

Gene expression ~ DNA methylation + Sample collection site + MGS + sex + age + 10 PCs + 15 SVs from the DNAm data and 13 SVs from the gene expression data

Multiple testing of CpGs per gene was corrected as described in QTLTools[45,46]. *P* values were calculated for the CpG regression coefficient, and empirical *p* values for the eQTM target genes were estimated using permutation analysis and beta distribution fit on the *p* values of all CpGs tested per gene[42,46]. Benjamini-Hochberg FDR procedure was applied to the eQTM empirical *p* values to account for the number of genes tested. Significant eQTMs, determined at FDR ≤ 0.05, were further considered for downstream analyses. Pearson's correlation coefficient was calculated for M-values and gene expression values for all significant eQTMs.

### Gene ontology and pathway enrichment of mQTL, eQTM and eQTL target genes

We used *GeneEnrich*[2] (https://github.com/segrelabgenomics/GeneEnrich) to identify gene ontology (GO) terms enriched for target genes of significant mQTL, eQTM and eQTL (FDR < 0.05). *GeneEnrich* estimates an empirical gene set enrichment *p* value for a list of significant genes in predefined gene sets, based on permutation analysis of a background set of genes expressed in a given tissue to adjust for potential biases that may arise from the genes expressed in the given tissue. Specifically, the gene set enrichment *P* value for a given gene set or pathway is computed as the fraction of 1000–100,000 randomly sampled sets of genes from a background (null) list of genes, of equal size to that of the significant set of genes, which have the same or more significant hypergeometric probability than the observed probability of the significant genes. The null set of genes was defined in this study as all genes expressed in peripheral retina that were tested for mQTLs, eQTMs and eQTLs, excluding the significant set of genes. Given the large number of significant m/eQTLs and eQTMs, we tested for gene set enrichment among the top 10% of mQTL, eQTL and eQTM target genes, ranked based on their gene level FDR values. Gene Ontology (GO) (http://geneontology.org/), Reactome, and Kyoto Encyclopedia of Genes and Genomes (KEGG) gene sets were downloaded from the Molecular Signature Database (MSigDB) (https://www.gsea-msigdb.org/gsea/msigdb/) on March 2021, which consisted of over 11,000 gene sets. Only the subset of genes in each gene set found in the given database and expressed in retina and tested for mQTLs, eQTMs and eQTLs, respectively, were considered in the analysis. We restricted the analysis to gene sets with a minimum of 10 genes to a maximum of 1000 genes after intersecting them with retina expressed genes. GO-Figure (v1.0.1)[88] (https://gitlab.com/evogenlab/GO-Figure) was used to identify and plot an non-redundant set of gene ontology terms that were significantly enriched for mQTL, eQTM and eQTL target genes based on gene membership overlap and *p* value ranking. Gene set enrichment significance was determined based on Benjamini-Hochberg FDR below 0.1 per database.

### Summary data-based mendelian randomization (SMR) analysis of eQTL, mQTL and AMD GWAS

We performed summary data-based mendelian randomization (SMR) analysis to test for causal or pleiotropic associations between eQTL and mQTL using eQTL summary statistics as the exposure and mQTL as the outcome (represented as E2M_SMR analysis), or between mQTL and eQTL using mQTL summary statistics as the exposure and eQTL summary statistics as the outcome (represented as M2E_SMR analysis), using package GSMR[40,41] (v1.1.1) (https://yanglab.westlake.edu.cn/software/gsmr/#Sourcecode). Three SNPs were used as the minimum number of instruments required for the analysis. For M2E_SMR analysis, we selected an LD-independent set of significant mQTL variants (*P* < 5E-05), irrespective of whether they are significant eQTLs, and vice versa for the E2M_SMR analysis, thereby abiding by the MR assumptions. We used the heterogeneity independent instruments

(HEIDI) test and *P* value threshold of greater than 0.05 to differentiate pleiotropy from linkage[25,40]. We also performed SMR and HEIDI analysis to test causal or pleiotropic associations between mQTL or eQTL and AMD GWAS. All analyses were applied to all significant independent variant-by-gene or variant-by-CpG site QTLs. Bonferroni correction was used to determine significance at α < 0.05.

## Colocalization analysis between retina mQTL or eQTL and known AMD GWAS loci

To identify retina mQTL and/or eQTL that might underlie the causal mechanisms of genetic associations with AMD in known GWAS loci, we applied a Bayesian-based colocalization method, eCAVIAR[42] (https://github.com/fhormoz/caviar) to each of the 52 independent AMD GWAS variants in 34 loci and overlapping peripheral retina mQTLs and eQTLs. Up to two independent causal variants per locus were assumed with eCAVIAR. Only mQTLs or eQTLs with at least 5 significant variants in the LD interval per GWAS locus ($r^2 > 0.1$ plus 50 kb on either side of the lead GWAS variant) were tested. All variants that fell within the GWAS locus LD interval were analyzed. A colocalization posterior probability (CLPP) above 0.01 was considered significant based on prior simulations[42]. To minimize false positive results, we excluded significant colocalizing GWAS-QTL signals if the m/eQTL and GWAS variants with CLPP > 0.01 did not pass the following significance cutoffs: m/eQTL FDR < 0.05 or $P < 10^{-4}$ and/or GWAS $P < 10^{-5}$. Supplementary Data 5 presents a summary of the colocalization results for 52 AMD GWAS loci. We added the direction of effect of the mQTL or eQTL on AMD risk if significant colocalization was found. LocusZoom (http://locuszoom.sph.umich.edu) and LocusCompare (https://github.com/boxiangliu/locuscomparer) plots were generated for the colocalization results.

## Colocalization analysis between eQTL, mQTL and AMD GWAS loci using Moloc

We performed Bayesian colocalization analysis to identify shared causal variants or haplotypes between co-occurring eQTL, mQTL, and AMD GWAS signals. To estimate the posterior probability that a lead variant is associated with two traits (GWAS and mQTL (GM), GWAS and eQTL (GE), mQTL and eQTL (EM)), coloc[43] (v5.1.1) (https://github.com/chr1swallace/coloc) was implemented using the R package. Next, to estimate posterior probability that a given variant is associated with three traits (GWAS, eQTL and mQTL (GEM)), moloc[44] (v0.10) (https://github.com/clagiamba/moloc) was implemented using the R package and the following priors: $p1 = 10^{-4}$, $p2 = 10^{-4}$, $p12 = 5 \times 10^{-6}$, allowing for up to 4 independent ($r^2 < 0.01$) causal variants per locus. Posterior probability above 0.8 was considered as evidence that two or three trait associations share the same causal variant based on coloc and moloc, respectively. The colocalization posterior probabilities output given by moloc account for the multiple pairwise comparisons performed[44]. LocusZoom (http://locuszoom.sph.umich.edu) and LocusCompare (https://github.com/boxiangliu/locuscomparer) plots were generated for colocalization results.

## Retina Hi-C and epigenetic data analyzed

Hi-C data, loops, *cis*-regulatory elements (CREs) and super-enhancers (SE) were taken from our previously published study of chromatin contact map of the adult human retina[36]. CREs in that study were identified by chromatin states enriched for active histone marks identified by ChromHMM v1.19 that were merged to generate a set of regions called CREs.

## Integration of Hi-C data for identification of target genes for mQTLs, eQTLs, eQTMs, variants and genes identified in SMR, eCAVIAR, coloc and moloc analyses

The mQTLs were classified based on the location of the variants relative to the canonical TSS of the associated mGene based on Illumina annotation[84]. eQTLs were classified based on the location of the variants relative to the canonical TSS of the associated eGene based on Ensembl 85 version. eQTMs were classified based on the location of the CpGs relative to the canonical TSS of the associated target genes based on Ensembl 85 version. mQTL and eQTL variants located within ± 2.5 kb of the mGene/eGene TSS were identified as promoter mQTLs/eQTLs and those located >2.5 kb from the mGene/eGene were identified as distal mQTLs/eQTLs. The mQTLs/eQTLs variants overlapping a chromatin loop foot were classified based on the location contacted by the opposite loop foot; variants in contact with their mGene/eGene promoter were identified as mGene/eGene promoter mQTLs/eQTLs, and variants in contact with a promoter other than the mGene/eGene were identified as non-eGene promoter mQTLs/eQTLs. The mQTLs/eQTLs were checked for overlap with CREs and SEs and subsequently for overlap with the associated mGene/eGene. eQTMs located within ± 2.5 kb of the TSS of their target gene were identified as promoter eQTMs, while those located >2.5 kb from the target gene were identified as distal eQTMs. eQTMs' target CpG which overlap a chromatin loop foot were classified based on the location contacted by the opposite loop foot; CpG in contact with their target gene promoter were identified as Gene promoter eQTMs, CpGs in contact with a promoter other than the gene were identified as non-Gene promoter eQTMs. eQTMs were checked for overlap with retina CREs and SEs and subsequently checked if those regions overlapped the associated target gene. All overlaps were performed using GenomicRanges (v1.42) (https://bioconductor.org/packages/release/bioc/html/GenomicRanges.html) in R. To confirm target genes for associations identified in SMR, colocalizations identified with eCAVIAR, coloc and moloc, and variants identified in these associations and colocalization analysis were overlapped with loops, CREs, and SEs. For a loop target gene, one foot of the loop overlaps the variant, and the second foot of the loop overlaps the gene body or TSS of a gene. CRE and SE target genes were defined by the variant and gene body (or TSS of a gene) overlapping the same CRE or SE. The closest target genes overlapping with loops, CREs, SE, mQTLs, eQTLs, eQTMs, SMR associations and colocalizations identified with eCAVIAR, coloc and moloc were obtained using the closestBed command from bedtools[89] (v2.27.1) (https://github.com/arq5x/bedtools2). The Hi-C data was visualized using Washington University Epigenome browser (http://epigenomegateway.wustl.edu/browser/).

## Reporting summary

Further information on research design is available in the Nature Portfolio Reporting Summary linked to this article.

# Data availability

The data that support this study are available from the corresponding author upon reasonable request. Dataset produced in this study are accessible in GEO under the accession number: GSE231536 (Methylation array). Gene expression data are from our previous study, under the accession GSE115828[30]. The summary statistics (nominal *p* values, beta and standard error) for all significant variant-CpG pair mQTLs, all significant variant-gene pair eQTLs, and all significant CpG-gene pair eQTMs are provided on Zenodo at the following URL: https://doi.org/10.5281/zenodo.10569726. Gene Ontology (GO) (http://geneontology.org/), Reactome, and Kyoto Encyclopedia of Genes and Genomes (KEGG) gene sets were downloaded from the Molecular Signature Database (MSigDB) (https://www.gsea-msigdb.org/gsea/msigdb/). The GWAS summary statistics for AMD are accessible under accession GCST00321926 [https://www.ebi.ac.uk/gwas/studies/GCST003219].

# Code availability

Publicly available software and packages were used throughout this study according to each developer's instructions. Custom scripts generated for the study will be available upon request.

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

## Acknowledgements

We thank N-NRL colleagues for their technical help and discussions, Andrew Singleton for support with DNA methylation, John M. Rouhana for providing details of RNA-seq QC. We thank Thomas Langmann (Laboratory for Experimental Immunology of the Eye, Department of Ophthalmology, University Hospital Cologne) and Armin Wolf (Department of Ophthalmology, University of Ulm) for collecting and providing donor eyes for the Cologne and Munich data set. We thank Andrea Milenkovic and Lisa Michaelis (Institute of Human Genetics, University of Regensburg) for sample processing, DNA extraction and quality assessment. Also, we are thankful to Markus Nöthen and Per Hoffmann from Life & Brain GmbH (Bonn, Germany) for performing the MethylationEPIC BeadChip array. This research was supported by Intramural Research Program of the U.S. National Eye Institute, (ZIAEY000450 and ZIAEY000546 to A.Sw.), NEI Extramural Research funding (R01 EY031424 to A.V.Se., P.A.M, and A.R.H., P30 EY014104 to S.M., and R01 EY028554 to D.A.F.), grants from Deutsche Forschungsgemeinschaft (DFG) (GR5065/1-1 to B.H.F.W.) and the Helen Lindsay Family Foundation (DAF). This study utilized the high-performance computational capabilities of the NIH Biowulf Linux cluster (http://biowulf.nih.gov). The funders had no role in study design, data collection and analysis, decision to publish, or preparation of the manuscript.

## Author contributions

Overall conceptualization: J.A., A.V.Se, and A.Sw.; clinical and tissue resources: D.A.F., S.R.Mo., B.H.F.W.; experimental work: F.V.A., D.G.H., R.R. and X.C.D.; bioinformatics analysis: J.A., P.A.M., A.R.H, S.Me., C.K., T.S., and M.K.; mQTL, eQTL and eQTM analysis: J.A.; data curation: J.A.; writing original draft: J.A., A.V.Se., and A.Sw.; writing, review, and editing: all authors; funding: E.Y.C., B.H.F.W., A.V.Se. and A.Sw.; supervision: B.H.F.W., A.V.Se. and A.Sw.; project administration: A.Sw.

## Funding

## Competing interests

The authors declare no competing interests.

## Additional information

[1]Neurobiology, Neurodegeneration and Repair Laboratory, National Eye Institute, National Institutes of Health, Bethesda, MD, USA. [2]Ocular Genomics Institute, Department of Ophthalmology, Massachusetts Eye and Ear, Boston, MA, USA. [3]Department of Ophthalmology, Harvard Medical School, Boston, MA, USA. [4]Broad Institute of MIT and Harvard, Cambridge, MA, USA. [5]Institute of Human Genetics, University of Regensburg, Regensburg, Germany. [6]Division of Epidemiology and Clinical Applications, Clinical Trials Branch, National Eye Institute, National Institutes of Health, Bethesda, MD, USA. [7]Laboratory of Neurogenetics, National Institute of Aging, National Institutes of Health, Bethesda, MD, USA. [8]Department of Ophthalmology and Visual Neurosciences, University of Minnesota, Minneapolis, MN, USA. [9]Institute of Clinical Human Genetics, University Hospital Regensburg, Regensburg, Germany. [10]Present address: Doheny Eye Institute, Pasadena, CA, USA. ✉e-mail: ayellet_segre@meei.harvard.edu; swaroopa@nei.nih.gov

