## [Peer Review File · Nature Communications]

QTL mapping of human retina DNA methylation identifies 87 gene-epigenome interactions in age-related macular degenerationEditorial Note: Parts of this Peer Review File have been redacted as indicated to remove third-party material where no permission to publish could be obtained.

REVIEWER COMMENTS

Reviewer #1 (Remarks to the Author):

The manuscript by Jayshree et al. presents a study on the associations of genotypes, DNA methylation, and gene expression in approximately 160 human retinas. The authors reported significant cis-mQTLs, eQTLs, and eQTM. Moreover, they conducted a comprehensive integrative analysis incorporating MR, colocalization, Hi-C, and pathway analysis. The study identified novel genes that exhibit potential correlations with age-related macular degeneration, thereby adding valuable insights to the field. The manuscript is well-written and effectively communicates the findings. However, I have the following comments to enhance the clarity and completeness of the work:

- 1) In the study conducted by Jayshree et al., the retinal samples included individuals with no AMD, intermediate AMD, and AMD samples. It would be beneficial if the authors could provide a description of the differentially methylated CpGs associated with AMD and compare these results with the findings from the MR analysis of AMD.
- 2) Since the retinal samples were derived from two independent studies, it is important to consider whether there might be any inflation of results due to merging the samples. Did the authors compare the results obtained from the two studies to assess potential discrepancies or similarities?
- 3) The authors used an FDR threshold of <0.05 for all their analyses. It would be helpful if they could also provide the corresponding p-values for the mQTL, eQTM, and eQTL analyses. Additionally, please clarify the method used for calculating the FDR. Was the Benjamini-Hochberg method employed?
- 4) It is recommended that the authors exclude DNA methylation probes that include SNPs. This step is crucial to the mQTL analysis and avoids any confounding effects introduced by SNPs.
- 5) The column names in Table 1 could be improved to enhance clarity. It would also be beneficial to include additional information about the CpGs, such as the distance between each CpG and its corresponding gene, as well as whether the CpG is located in the promoter, enhancer, or gene body region.
- 6) The authors conducted E2M and M2E MR analyses. However, there is some confusion regarding whether these tests violate the assumptions of MR analysis. Many cis-pairs of CpGs and genes actually share the same instrumental variables (SNPs), which might affect the validity of the MR tests. Further clarification on this matter would be appreciated.
- 7) In the SMR analysis, it is not clear whether the authors used a single SNP as the instrumental variable for each test or multiple SNPs. It would be helpful to provide a clear explanation of the methodology employed in this regard.

Reviewer #2 (Remarks to the Author):

Summary

Advani et al. present an interesting and intriguing body of work exploring the relationship between genetic variation, DNA methylation, and gene expression in the human retina. Their findings have a clear impact on our understanding of the regulation of gene expression in the retina, and a potential impact on our understanding of Age-related Macular Degeneration. I have some concerns and suggestions outlined below.

- 1) One of the primary concerns for this study is that the retinal tissue used for both the expression and the DNA methylation is from the peripheral retina. While this is fine for the standalone study of retina eQTL, eQTM, and methylationQTL, it is not clear that any of the findings are translatable to Age-related macular degeneration, which happens in the macula area of the retina. Multiple studies have shown profound differences in macular versus peripheral ocular tissues in their: 1) eQTL, 2) gene expression, and 3) cellular composition. To make the claim that these findings are meaningful for

macular degeneration, the authors would need to show that their findings are conserved in both the macula and the periphery, or alternatively remove the claims that these findings are applicable to AMD.

2) A second major concern is related to the post-mortem interval. From a reading of the methodology on the tissue processing, it sounds like the time between death and when the tissue is processed and the samples were frozen is likely too long. The authors refer to a 4-6hr time between death and enucleation. However, the relevant metric/interval that is currently not indicated is the time between death of the donors, to when the processed samples were placed in the freezer. The reason to stress this point is that we know that RNA is rapidly degraded, and the findings reported could be compromised by this time interval. I am aware that some scientists claim that the retina is an immune-privileged tissue and that therefore those time intervals do not apply to them, but this is not a valid argument, since: 1) the retina is not immune-privileged, 2) the retina/blood barrier is compromised in AMD, and 3) immune-privilege is independent RNase activity upon death. I would like to think that the findings would be the same in samples with a time interval of 4-8hours, but we do not know that. Have the authors examined gene expression, or eQTL, eQTM etc, in retinal tissues with different time intervals between death and sample freezing? What did the sample quality look like when placed in the context of this interval? Did the authors take this into account for the modeling of the eQTL, eQTM, or any of the downstream analyses? I do not see that in the methods.

3) A third major concern is the lack of validation for any of the findings presented. Experimental validation of one or more of the causal relationships between DNA methylation and expression, or these and AMD disease biology would substantially strengthen the manuscript, and their claims of the translatability of these findings to AMD.

4) Were CpGs that are also C->T SNPs excluded from the analysis? These should be excluded from methylation analyses, since the T cannot be methylated and hence the findings for those may be independent of methylation, and may instead related to the genetic variation.

5) Did the authors collapse CpGs into CpG islands or regions? Are the findings reported driven by singleton CpGs or by CpG islands?

6) Have the authors compared their QTL results to a QTL approach that uses a linear mixed model to robustly account for population structure in the study population? For example using FastLMM, pyLMM, or Rqt12 ? Some of these would also allow you to use a continuous predictor such as methylation, or you could binarize the DNA methylation data.

7) Did the authors employ the permutation based approach to estimate false discovery for eQTL and mQTL (via QTLtools/FastQTL)? I could not find the FDR approach for these two in the methods. If so, why did the authors use the less robust Benjamini-Hochberg approach to estimate FDR in the eQTM analysis? The authors should use a permutation based approach to estimate FDR for all their QTL and QTM analyses.

8) The authors' finding that roughly 45% of eQTM genes showed a positive correlation between expression and DNA methylation is intriguing. In the methylation field, it has been observed that such a positive correlation coincides with open chromatin, and hypothesized that this makes the DNA more accessible to DNA methyl transferases. It would be interesting to examine chromatin accessibility in the retina for the genes that show these positive correlation.

9) For the loci where methylation was predicted to have a causal effect on the gene expression in the SMR analyses, was the expression of all these genes inversely correlated with DNA methylation? What did that look like across all those genes?

10) The authors should clarify the rationale for using various mendelian randomization and colocalization approaches: SMR, Coloc, eCAVIAR, Moloc. Why is more than one needed, how they complement each other, and how consistent (or inconsistent) were the findings across them. These sections read a bit like a laundry list of analyses so it would be important to put them in context. For example, did the authors use both SMR and coloc as two methods to independently identify causal relationships between methylation and gene expression? ie. where the reader would have higher confidence in genes identified by both methods. Or, are both results needed to give confidence to any gene? The authors should clarify.

11) The authors used multiple pairwise comparisons and multiple tools for the various colocalization analyses. Did the authors correct for multiple testing across all the pair-wise tests?

12) There is no methods section for HiC data or for the CRE. It is difficult to properly evaluate the HiC data and results without its method section. How were CREs defined? How were the CRE identified?

13) From reading methods section on integration of HiC with other data types, it sounds like the integration is essentially overlap between the different xQTL/M using Genomics Ranges. Is this correct?

Minor comments

14) QTLs should be changed to QTL throughout the manuscript. The L=Locus (singular) or L=Loci (plural). QTLs is incorrect.

15) How are "target genes" defined in this paper? I would have guessed a target gene was the gene whose expression was mapped to the eQTM, but I also saw "mQTL target genes" (line 215). Does this refer to the hypothetical effect of mQTL on nearby genes? There are also "eQTM target genes" (Line 260). Please define these clearly.

16) The authors define mQTL, eQTL, eQTM in the Results section but not in the Methods section. These should also be defined in the methodology. Others are not clearly defined anywhere, for example "mVariants".

17) Sometimes the authors say meQTL instead of mQTL (ex Figure 1F, or in Suppl Figure Legend 2F). Edit to be consistent throughout the manuscript.

18) I would suggest to work on the figure labels. For example, the axes on Figure 1B (and multiple similar figures) are confusing. There are two axes, and 3 plots (2 bars and one line), which one goes with which? Similarly, 2F and 2G need axes labels, I see these are chromosomes on the legend, but the axes should be labeled on the figure.

19) Figure 5A, inset with GE, should "QTL" read "eQTL"?

20) Figure 5C, I would advise against using a circle plot. Honestly it is very hard to read it, or see where the lines fall. Even multiple manhattan plots would be better.

Reviewer #3 (Remarks to the Author):

This is a very well-described analysis of human retina tissue integrating genetic, transcriptional, and epigenetic analyses to identify genes and pathways regulated under normal conditions that may also be influenced by advancing age and/or environmental factors.

The authors present a great deal of information to support their analyses. The results were very important, but also very long, and I would find it incredibly helpful to have a bit more interpretation in the discussion of why each of these analyses were done and how they contribute to the overall findings/message of the paper and to the field.

I have a few additional questions and suggestions for the authors:

Lines 178-179: "749,158 CpG sites that passed stringent QC criteria (see Methods) using QTLtools42." -- the stringent QC criteria are not found in the methods.

Lines 229-232: "To evaluate the tissue-specificity of DNAm and mQTLs, we compared 36,906 methylated CpGs with significant mQTLs and 37,453 independent mQTLs in the retina to those identified in five different tissues: adipose (n=119)45, blood (n=614)46, endometrium (n=66)47, brain frontal cortex (n=526)48, and skeletal muscle (n=282)20." -- Please explain the how and why the comparison tissues of choice were determined.

Lines 250-252: "Top 10 genotype PCs, known covariates (e.g., AMD grade), and SVs capturing hidden covariates of expression, were included in the linear regression model (see Methods)." -- I do not see any elaboration in the methods of the "known covariates" and would find it helpful to know what those were.

Line 275: "on a few smaller chromosomes" -- please explain somewhere in the manuscript why the chromosome size is important for this type of analysis.

Given the mention of the effects smoking, exercise, and diet have on epigenetics, and the known influence of these factors on AMD, did the authors have smoking and/or diet data on the retina donors and did they consider utilizing this in analyses in any way?

To appeal to readers who are less familiar with these types of analyses, how they are interpreted, and why they are important, a concise concluding statement/section on the implications of these findings would be helpful.

Response to Reviewers' Comments

Reviewer #1 (Remarks to the Author):

The manuscript by Jayshree et al. presents a study on the associations of genotypes, DNA methylation, and gene expression in approximately 160 human retinas. The authors reported significant cis-mQTLs, eQTLs, and eQTMs. Moreover, they conducted a comprehensive integrative analysis incorporating MR, colocalization, Hi-C, and pathway analysis. The study identified novel genes that exhibit potential correlations with age-related macular degeneration, thereby adding valuable insights to the field. The manuscript is well-written and effectively communicates the findings. However, I have the following comments to enhance the clarity and completeness of the work:

1) In the study conducted by Jayshree et al., the retinal samples included individuals with no AMD, intermediate AMD, and AMD samples. It would be beneficial if the authors could provide a description of the differentially methylated CpGs associated with AMD and compare these results with the findings from the MR analysis of AMD.

We thank the reviewer for this suggestion. We carried out differential methylated CpG analysis in retina between no AMD (MGS1) vs. intermediate AMD (MGS3) and no AMD (MGS1) vs. advanced AMD (MGS4). We could identify only one significant differential CpG using Benjamini-Hochberg $FDR \leq 0.05$ for the MGS1 vs. MGS3 comparison, and no significant CpGs for MGS1 vs. MGS4 at $FDR \leq 0.05$. This is in concordance with the findings other groups identifying only a few differentially methylated CpGs between control samples and early, intermediate and advanced AMD in macula RPE/choroid tissue (see Orzoco et al., Cell Genomics 2023; PMID: 37388919), and in blood samples of patients with and without AMD (PMID: 26067391). It is possible that we are underpowered to detect differential methylation with AMD grade at our current sample size. In the future, we plan to revisit this analysis with larger sample sizes.

We added a few sentences describing this analysis in the Methods section on page 31, lines 733-737 and point to this in the Results section on page 9, lines 181-183, noting that AMD grade of the samples does not have an effect on the mQTL analysis.

2) Since the retinal samples were derived from two independent studies, it is important to consider whether there might be any inflation of results due to merging the samples. Did the authors compare the results obtained from the two studies to assess potential discrepancies or similarities?

We are grateful to you for raising this important point. We did not detect, in Supplementary Figure 1a and 1b, any significant population difference between the principal component analysis (PCA) of the genotype data of samples from the USA (Bethesda) and Germany (Cologne and Munich). Nevertheless, we added the top 10 genotype PCs to the mQTL regression model to ensure that subtle population variation does not affect the mQTL analysis results. Furthermore, while PCA of the CpG M-values across all samples showed a batch effect of CpG measurements between the studies, this effect was removed when regressing out 38 surrogate variables (SVs) from the M-values as shown in Supplementary Figure 1d vs. 1c. To correct for potential batch effects in the mQTL analysis, we adjusted for sample collection site, disease status (MGS level), age, sex, population stratification (top 10 genotype PCs), and 15 SVs in our mQTL mapping, since 15 SVs maximized the number of CpG sites with significant mQTLs and the number of mQTLs at

FDR \leq 0.05. We describe this in the Methods section on page 31, lines 747-750, and added a panel in Supplementary Figure 1e showing that 15 SVs also correct for the batch effect on the M-values.

To further assess potential inflation of results due to merging of samples from two studies, we performed mQTL analysis on each of the two studies separately (96 RNA-seq samples from the Bethesda study and 56 RNA-seq samples the Cologne and Munich (Germany) study), as suggested by the reviewer, correcting for the same set of covariates. In comparing the results between the two studies, we detected 86,053 mQTL variants at FDR $<$ 0.05 (74% of the Germany study mQTLs) overlapping between the two studies, and 2,045 CpGs at FDR $<$ 0.05 having at least one significant mQTL in both studies (86% of the CpGs with significant mQTLs in the Germany study). Approximately 80% of the mQTLs from Germany study were present in the combined mega-analysis study, and 56% of the Bethesda study mQTLs were observed in the combined mega-analysis study. 68% of the mQTLs and 78% of the CpGs with significant mQTLs in the mega-analysis were identified in the larger Bethesda study. This is not surprising as we expect to observe differences due to sample sizes. Given these results and those we show in Supplementary Fig. 1 to correct for potential batch effects, we believe that mega-analysis is a reasonable approach and is likely not confounded by inflation. Additionally, of note, the number of CpG sites (36,906) for which we found significant mQTLs at FDR $<$ 0.05 with the mega-analysis is rather conservative relative to the number of CpGs with significant mQTLs (on the order of 100,000) discovered in nine GTEx tissues with similar sample size (see Fig. 2a in Oliver *et al.* Nat Genet 2022, PMID: 36510025).

3) The authors used an FDR threshold of $<$ 0.05 for all their analyses. It would be helpful if they could also provide the corresponding p-values for the mQTL, eQTM, and eQTL analyses. Additionally, please clarify the method used for calculating the FDR. Was the Benjamini-Hochberg method employed?

We thank the reviewer for the opportunity to clarify these points. We have provided the corresponding p-values, beta and standard error for all independent significant (FDR $<$ 0.05) variant by CpG, variant by gene, or CpG by gene pair signals for the mQTL, eQTL and eQTM analyses in Supplementary Table 2, Supplementary Table 3, and Supplementary Table 8, respectively. We also now uploaded the summary statistics (nominal p-values, beta and standard error) for all significant variant-CpG pair mQTLs, all significant variant-gene pair eQTLs, and all significant CpG-gene pair eQTMs on Zenodo at the following URL: <https://doi.org/10.5281/zenodo.10246444> . We added a sentence stating this in Data Availability on page 40.

Two levels of multiple hypothesis correction were applied in the mQTL, eQTL and eQTM analyses. First, the number of variants tested per CpG or gene or number of CpGs tested per gene were adjusted for using permutation analysis and a fitted beta-distribution, yielding empirical p-values per CpG or gene tested. To account for the number of CpGs tested in the mQTL analysis and number of genes tested in the eQTL analysis the Storey & Tibshirani False Discovery Rate (FDR) procedure, implemented in the QTLtools software suite, was applied to the empirical p-values. For eQTM analysis, we used the Benjamini-Hochberg False Discovery Rate to correct for number of genes tested, which was a bit more stringent than Storey's FDR correction (13,747 eQTM genes were significant at BH FDR $<$ 0.05 vs. 17,367 eQTM genes at Storey's FDR $<$ 0.05), as applied in two recent eQTM studies in skeletal muscle (PMID: 31076557) and kidney (PMID: 35710981). Significance was determined at a threshold of FDR \leq 0.05. We added these details in the manuscript in the Methods section on pages 31 (lines 750-755), 34 (lines 807-811) and 35 (lines 849-854).

4) It is recommended that the authors exclude DNA methylation probes that include SNPs. This step is crucial to the mQTL analysis and avoids any confounding effects introduced by SNPs.

We agree with the reviewer and apologize for not stating this clearly. We had indeed excluded the DNA methylation probes that included SNPs before doing the mQTL analysis. We have now added this detail in the manuscript on page 30, line 717.

5) The column names in Table 1 could be improved to enhance clarity. It would also be beneficial to include additional information about the CpGs, such as the distance between each CpG and its corresponding gene, as well as whether the CpG is located in the promoter, enhancer, or gene body region.

We thank the reviewer for the suggestion. We have clarified the Table 1 headers and added additional columns about the CpGs, including distance between each CpG and its corresponding gene transcription start site (TSS) and location of the CpG relative to the gene in Table 1 on page 43. We also included a few clarifying descriptions in the footnote of the table.

6) The authors conducted E2M and M2E MR analyses. However, there is some confusion regarding whether these tests violate the assumptions of MR analysis. Many cis-pairs of CpGs and genes actually share the same instrumental variables (SNPs), which might affect the validity of the MR tests. Further clarification on this matter would be appreciated.

We appreciate the opportunity to clarify this point. We performed summary statistics-based Mendelian randomization (SMR) analysis with retina eQTL and mQTL summary statistics considering either CpG methylation (DNAm) as the exposure and gene expression as the outcome (represented as M2E_SMR) or gene expression as the exposure and DNAm as the outcome (represented as E2M_SMR). In each case, we selected the instrumental variables (IVs) only based on the variants being significantly associated ($P < 5E-05$) with the QTL type that was the exposure, blinded by the significance of the other QTL type (the outcome). In case of M2E_SMR, we selected an LD-independent set of significant mQTL variants, agnostic to whether they are significant eQTLs, and vice versa for the E2M_SMR analysis. Thus, while there is some overlap between the significant mQTL and eQTL variants (40% of the mVariants are significant eVariants and 45% of the eVariants are mVariants at $FDR < 0.05$), this does not violate the assumptions of MR analysis. We added a sentence on this in the Methods section on page 37, lines 887-891.

We only detected a small fraction of variants to be significant mQTLs and eQTLs, as stated on page 14, lines 319-320. The results are summarized on page 14, lines 310-320: "In the M2E_SMR analysis, 9,307 associations (SMR P-value $< 9.38 \times 10^{-7}$ after Bonferroni correction) (Supplementary Table 14) were evident, of which 6,592 associations passed the linkage test (HEIDI P-value > 0.05) corresponding to 5,175 mQTLs, 5,012 eQTLs and 2,101 genes (Fig. 3d; 51% of the M2E_SMR QTL effect sizes were negatively correlated and 49% positively correlated). Of the 6,592 M2E_SMR associations, 232 associations (3.5%) corresponding to 232 genes were also identified as significant eQTLs of which 176 (75.8%) eQTLs showed negative correlation and 56 (24.1%) eQTLs revealed positive. We detected 554 common associations between E2M_SMR and M2E_SMR (Supplementary Fig. 5a), proposing a feedback mechanism between DNAm and gene expression regulation for a minority of the 513 mQTLs (1.3%) and 292 eQTLs (2.3%). A larger number of significant M2E_SMR associations compared to E2M_SMR suggests that the genetic effect of DNAm on gene expression is a more predominant mechanism than the genetic effect of gene expression on DNAm."

7) In the SMR analysis, it is not clear whether the authors used a single SNP as the instrumental variable for each test or multiple SNPs. It would be helpful to provide a clear explanation of the methodology employed in this regard.

Thank you for the question. We used 3 SNPs as the minimum number of instrumental variables required for each test. This enables us to test for horizontal pleiotropy that requires at least 3 variants for all cases. We have added this detail in the Methods section on page 37, lines 887-888.

Reviewer #2 (Remarks to the Author):

Summary

Advani et al. present an interesting and intriguing body of work exploring the relationship between genetic variation, DNA methylation, and gene expression in the human retina. Their findings have a clear impact on our understanding of the regulation of gene expression in the retina, and a potential impact on our understanding of Age-related Macular Degeneration. I have some concerns and suggestions outlined below.

1) One of the primary concerns for this study is that the retinal tissue used for both the expression and the DNA methylation is from the peripheral retina. While this is fine for the standalone study of retina eQTL, eQTM, and methylationQTL, it is not clear that any of the findings are translatable to Age-related macular degeneration, which happens in the macula area of the retina. Multiple studies have shown profound differences in macular versus peripheral ocular tissues in their: 1) eQTL, 2) gene expression, and 3) cellular composition. To make the claim that these findings are meaningful for macular degeneration, the authors would need to show that their findings are conserved in both the macula and the periphery, or alternatively remove the claims that these findings are applicable to AMD.

We apologize for the oversight. We should have explained this important point in our previous version. AMD is a pan-retinal disease and not just a disease of the macula (though the central vision loss is due to death of the macular RPE and photoreceptors). In fact, the lesions go way out to the peripheral retina. Panretinal cone dysfunction has been observed in AMD patients (PMID: 17471344). In AREDS2 study, it has been demonstrated that persons with intermediate AMD, 97% of them have drusen, geographic atrophy and neovascular process that goes far into the retinal periphery (PMID: 28089680). Having said that, AMD also affects preferentially the rod function, and these are dense in the parafoveal area and in the retinal periphery. Curcio *et al.* have published a paper on rods and the effect of AMD on rods and the distance from the center of the fovea (PMID: 11450761). We added a sentence summarizing these points in the Introduction on page 7, lines 121-122. Furthermore, a study that analyzed bulk RNA-seq data of human retina showed differential expression of a relatively small number of genes (754) between macula and peripheral retina (Orzoco *et al.*, Cell Reports 2020, PMID: 31995762), suggesting that the peripheral retina also captures molecular changes relevant for AMD pathology.

2) A second major concern is related to the post-mortem interval. From a reading of the methodology on the tissue processing, it sounds like the time between death and when the tissue is processed and the samples were frozen is likely too long. The authors refer to a 4-6hr time between death and enucleation. However, the relevant metric/interval that is currently not indicated is the time between death of the donors, to when the processed samples were placed in the freezer. The reason to stress this point is that we know that RNA is rapidly degraded, and

the findings reported could be compromised by this time interval. I am aware that some scientists claim that the retina is an immune-privileged tissue and that therefore those time intervals do not apply to them, but this is not a valid argument, since: 1) the retina is not immune-privileged, 2) the retina/blood barrier is compromised in AMD, and 3) immune-privilege is independent RNase activity upon death. I would like to think that the findings would be the same in samples with a time interval of 4-8hours, but we do not know that. Have the authors examined gene expression, or eQTL, eQTM etc, in retinal tissues with different time intervals between death and sample freezing? What did the sample quality look like when placed in the context of this interval? Did the authors take this into account for the modeling of the eQTL, eQTM, or any of the downstream analyses? I do not see that in the methods.

We thank the reviewer for raising this valid concern. Tissues were collected over a period of almost 20 years. It is extremely hard to get retina samples in few hours when collecting hundreds of samples. We have previously published an RNA-seq and eQTL study with the same donor samples (406) and considered only the samples for RNA-seq and eQTL analysis having RIN ≥ 6 (Ratnapriya *et al.*, 2019; PMID: 30742112). In this study, we described the distribution of postmortem interval (PMI), which ranged primarily between 5 and 24 hours, the RNA quality (RIN) for these donors, and correlation between RIN number and postmortem interval in Supplementary Figure 1 d, e, f (see below). The negative correlation between RIN and PMI was small (Pearson correlation coefficient, $r = -0.0935$, $R^2=0.0087$), suggesting a minor effect of PMI on RNA quality. With that, we are correcting for PMI and RIN numbers in the eQTL and eQTM analyses in our manuscript by adding surrogate variables (SVs; inferred hidden covariates of gene expression) that capture gene expression variation due to PMI (see correlation results between PMI and RIN and the SVs in Supplementary Figure 2c in Ratnapriya *et al.*, 2019 and below) to the regression models, as done in Ratnapriya *et al.*, 2019.

This is standard practice for QTL analyses, which was also used in GTEx study (GTEx consortium, Science 2020; GTEx mQTL paper NG 2023). We added a sentence on the PMI and how it was adjusted for in the eQTL and eQTM models in the Methods section on page 28, lines 663-665. Additionally, there are steps taken at the hospital (donor eyes have ice packs placed on them shortly after death) and then in transport (on an ice pack in a cooler) that slows down the activity of degradative enzymes. We should note that RNase is relatively less in the neural retina compared to other tissues such as pancreas and liver where RNase content is a huge problem.

Supplementary Figure 1 d, e, f taken from Ratnapriya *et al.*, Nature Genetics 2019:

[redacted]

Supplementary Figure 2c taken from Ratnapriya *et al.*, Nature Genetics 2019:

[redacted]

3) A third major concern is the lack of validation for any of the findings presented. Experimental validation of one or more of the causal relationships between DNA methylation and expression, or these and AMD disease biology would substantially strengthen the manuscript, and their claims of the translatability of these findings to AMD.

We thank the reviewer for the advice. We are planning to do these experiments for validation of the causal relationships between DNA methylation and gene expression in the context of AMD disease biology. However, the changes in effects for the validation of these casual relationships would likely be small, and no good model system is available for AMD. We are currently designing such follow-up studies, which will require considerable time and are beyond the scope of this manuscript.

We note that DNA methylation changes are suggested to correlate with aging and lifestyle factors (PMID: 22215131, PMID: 35501397). In our data, we are uncovering some relevant pathways, such as the glutathione metabolism, glycolysis, and immune response. We are targeting specific experiments to test the effect of DNA methylation and expression on these processes. In future studies, whole-genome bisulfite sequencing and single-cell DNA methylation profiling on larger cohort size can also contribute to address these casual relationships.

4) Were CpGs that are also C->T SNPs excluded from the analysis? These should be excluded from methylation analyses, since the T cannot be methylated and hence the findings for those may be independent of methylation, and may instead related to the genetic variation.

Thank you for pointing this out. We have indeed excluded the DNA methylation probes that include C->T SNPs before doing the mQTL analysis described in our submitted manuscript version. We have now added these details in the manuscript on page 30, line 717.

5) Did the authors collapse CpGs into CpG islands or regions? Are the findings reported driven by singleton CpGs or by CpG islands?

Thank you for helping us clarify this point. We have reported all the results based on single CpGs and now clarify this point on page 32, line 757.

6) Have the authors compared their QTL results to a QTL approach that uses a linear mixed model to robustly account for population structure in the study population? For example using FastLMM, pyLMM, or Rqt12 ? Some of these would also allow you to use a continuous predictor such as methylation, or you could binarize the DNA methylation data.

We thank the reviewer for the suggestion. Mixed effects models are useful for studies with complex populations structure, such as biobank studies. Given that we removed related individuals from our sample set and the ancestral background is relatively homogeneous (see Supplementary Figure 1a and 1b), we do not believe that we need to use a linear mixed model in our mQTL study. We are correcting for population structure by adding the top genotype PCs in our QTL analysis. The mixed effects model could also reduce statistical power (PMID: 24473328). In similar studies as ours, the linear regression model with a fixed effect, implemented in QTLtools and FastQTL, is a standard approach for eQTL and mQTL analyses, as performed in GTEx (GTEx consortium, Science 2020; PMID: 32913098 and GTEx mQTL paper Nature Genetics 2023; PMID: 36510025) and other studies (PMID: 31076557, 35710981).

7) Did the authors employ the permutation based approach to estimate false discovery for eQTL and mQTL (via QTLtools/FastQTL)? I could not find the FDR approach for these two in the methods. If so, why did the authors use the less robust Benjamini-Hochberg approach to estimate FDR in the eQTM analysis? The authors should use a permutation based approach to estimate FDR for all their QTL and QTM analyses.

Thank you for raising this question. Permutation-based approach was indeed used to estimate false discovery rate for eQTL and mQTL using QTLtools. First, the number of variants tested per CpG, or gene were adjusted for using permutation analysis and a fitted beta-distribution on the genotype regression coefficient p-values yielding empirical p-values per CpG or gene tested, as described in QTLtools. Storey & Tibshirani False Discovery Rate (FDR) procedure was then applied to these empirical p-values to account for number of CpGs tested in the mQTL analysis or number of genes tested in the eQTL analysis, as implemented in the QTLtools software suite. Significance was determined at a threshold of $FDR \leq 0.05$.

For the eQTMs, we also used permutation analysis to estimate the eQTM significance. Empirical P-values for eQTMs were calculated on the CpG regression coefficients correcting for all CpGs tested per gene using a beta distribution fit with permutations. We used the Benjamini-Hochberg (BH) False Discovery Rate (FDR) procedure to correct for number of genes tested in the eQTMs analysis, as it was a bit more stringent than Storey's FDR correction in our case (13,747 eQTM genes were significant with BH $FDR < 0.05$ and 17,367 with Storey's $FDR < 0.05$), and also because BH FDR correction was applied in two recent eQTM studies in skeletal muscle (PMID: 31076557) and kidney (PMID: 35710981). Significance was determined at $FDR \leq 0.05$.

We now added the details of the multiple hypothesis correction applied in the mQTL, eQTL and eQTM analyses in our Methods section on pages 31, 34 and 35.

8) The authors' finding that roughly 45% of eQTM genes showed a positive correlation between expression and DNA methylation is intriguing. In the methylation field, it has been observed that such a positive correlation coincides with open chromatin, and hypothesized that this makes the DNA more accessible to DNA methyl transferases. It would be interesting to examine chromatin accessibility in the retina for the genes that show these positive correlation.

We thank the reviewer for this interesting suggestion. We took chromatin accessibility (ATAC-Seq) data from our previously published study on adult human retina (PMID: 36207300) that

recorded open chromatin regions observed in at least 3 out of the 5 samples. We examined chromatin accessibility footprints for 6,267 genes (45%) that are target genes to positively correlated eQTM and observed 5,057 genes (80% of the target genes) overlapping an open chromatin region in retina. We also evaluated the genes with negative correlation of DNAm and gene expression and found a slightly lower overlap with open chromatin regions, which was not statistically significantly different than the overlap for the positively correlated target genes. This suggests that chromatin accessibility is likely not a predictor of the direction of effect of eQTMs. We added a description of this analysis on page 13, lines 274-281.

9) For the loci where methylation was predicted to have a causal effect on the gene expression in the SMR analyses, was the expression of all these genes inversely correlated with DNA methylation? What did that look like across all those genes?

Thank you for this important question. We have previously addressed these points with respect to the eQTMs in the Results section on page 14, lines 315-316. We now also added the percentage of inverse correlation between the mQTLs predicted to affect gene expression from the SMR analysis and the eQTM subset to the sentence below on page 14, lines 310-322 in the manuscript (highlighted in yellow):

“In M2E_SMR analysis where methylation was predicted to have a causal effect on the gene expression in the SMR analysis, we identified 9,307 associations (SMR P-value < 9.38×10^{-7} after Bonferroni correction) (Supplementary Table 14) of which 6,592 associations passed the linkage test (HEIDI P-value > 0.05) corresponding to 5,175 mQTLs, 5,012 eQTLs and 2,101 genes (Fig. 3d; 51% of the M2E_SMR QTL effect sizes were negatively correlated and 49% positively correlated). Of the 6,592 M2E_SMR associations, 232 associations (3.5%) corresponding to 232 genes were also identified as significant eQTMs, of which 176 (75.8%) eQTMs showed negative correlation and 56 (24.1%) eQTMs revealed positive correlation.”

10) The authors should clarify the rationale for using various mendelian randomization and colocalization approaches: SMR, Coloc, eCAVIAR, Moloc. Why is more than one needed, how they complement each other, and how consistent (or inconsistent) were the findings across them. These sections read a bit like a laundry list of analyses so it would be important to put them in context. For example, did the authors use both SMR and coloc as two methods to independently identify causal relationships between methylation and gene expression? ie. where the reader would have higher confidence in genes identified by both methods. Or, are both results needed to give confidence to any gene? The authors should clarify.

We appreciate the opportunity to clarify the rationale for applying multiple colocalization and causal inference approaches in our paper. We applied several colocalization methods, including coloc, eCAVIAR, and moloc and a Mendelian randomization (MR) method to cast a wide net of discovery of casual relationships between DNA methylation (mQTL) and gene expression (eQTL) QTLs and between mQTL, eQTL and AMD GWAS loci. This is because as discussed in the review Zuber *et al.*, AJHG 2022 (PMID: 35452592), each method has its limitations and strengths, and colocalization and MR methods complement each other. Colocalization methods (coloc, eCAVIAR, and moloc) test the hypothesis that co-occurring molecular QTLs and/or GWAS loci share the same causal variant, and if so, this suggests that the molecular QTL may be causal to the phenotype/disease association. On the other hand, MR directly tests for a causal relationship between QTLs and GWAS associations, and the summary data-based MR (SMR) method we applied also tests whether the significant causal is due to a causal or pleiotropic relationship versus linkage of two distinct causal variants.

We applied several colocalization methods due to their differences in underlying assumptions: coloc assumes a single causal variant per locus and is more powerful to detect colocalization for such scenarios, while eCAVIAR and moloc can assume more than one causal variant per locus (allelic heterogeneity), which is important since a substantial amount of QTLs and GWAS loci have been shown to have more than one underlying causal variant/haplotype. moloc allows to simultaneously test for sharing of causal variant/s between three molecular and trait associations, such as eQTLs, mQTLs and GWAS loci, as applied in our paper, while coloc, eCAVIAR, and SMR can only be applied to two traits at a time. The overlap of results between methods was limited, as described on pages 15-19 in results section “Causal or pleiotropic relationships between genetic regulation of DNAm and gene expression”, lines 342-343 “SMR and colocalization of retina mQTLs and eQTLs prioritize causal genes for AMD loci”, lines 391-394 “Colocalization analysis of retina mQTLs, eQTLs and GWAS identifies additional AMD genes”, lines 420-424 and lines 430-436. We further applied coloc and moloc to all significant eQTLs and mQTLs genome-wide in relation to AMD GWAS to also discover new AMD associations that may be driven by eQTLs and/or mQTL (results described on page 17/18).

Due to the differences between method, a limited number of loci were found to be significant with more than one method. We thus used a unique union across all methods to prioritize a comprehensive list of putative causal relationships between mQTLs and eQTLs, and between mQTLs, eQTLs and AMD associations, and to propose candidate genetic and epigenetic changes and causal genes for AMD for follow-up functional studies, as noted at the end of the Results section on page 21/22:

“To obtain a non-redundant set of genes, we merged the target genes identified in mQTL and AMD GWAS analyses using SMR, eCAVIAR, coloc and moloc. We identified 4 target genes at 3 loci by SMR, 15 target genes at 8 loci using eCAVIAR, 58 target genes at 11 loci using coloc, and 10 target genes at 3 loci using moloc. By combining the target genes from the four methods, we identified a total of 87 non-redundant genes that are influenced by DNAm (Supplementary Table 39). Fifty of these genes fall in reported AMD GWAS loci and 37 are target genes in potentially new AMD associations.”

Results with more than one method were proposed as a high confidence set of causal relationships, which we noted for each method on pages 15-19. We also used Hi-C loops to further provide support for a causal relationship between mQTLs and eQTLs and AMD loci, as we noted in the Discussion on page 23. Integration with retina Hi-C data has yielded supportive evidence of regulatory mechanisms and helped in determining high confidence target genes for mQTLs and eQTLs.

We clarified the usage of various colocalization and MR methods in the Results and discussion sections and added Table 2 that summarizes the significant findings with each method for the different combinations of mQTL, eQTL and AMD GWAS loci. In the Results under the ‘Overview of the analysis workflow’ section on top of page 9, we made the following edits in yellow: “To **provide a comprehensive set of** causal relationships between *cis*-mQTLs and *cis*-eQTLs, and between m/eQTLs and AMD GWAS signals, we pursued two **complementary** approaches **with different underlying assumptions**: (i) SMR to distinguish pleiotropic or causal association from linkage of genetic associations with DNAm, gene expression and AMD^{37,19,38}, and (ii) colocalization analyses that test whether co-occurring association signals are tagging the same causal variant/haplotype, including eCAVIAR³⁹ (**assuming up to two causal variants**), coloc⁴⁰ (**assuming a single causal variant**), and multiple-trait-coloc (moloc)⁴¹ (**assuming up to four causal variants**)”.

We have revised the Discussion to explain why we applied multiple methods and its advantages and limitations, including on page 22: “We support this proposal by identifying mQTLs that may be causal to eQTLs and vice versa using Mendelian randomization and colocalization analyses. The strength of our study is that we applied two different complementary approaches (Zuber *et al.*, AJHG 2022, PMID: 35452592) to testing whether mQTLs, eQTLs and/or AMD associations share a common causal variant and whether their association is causal or pleiotropic as opposed to being confounded by linkage of distinct causal variants.” and expanded on this point on page 25.

11) The authors used multiple pairwise comparisons and multiple tools for the various colocalization analyses. Did the authors correct for multiple testing across all the pair-wise tests?

We thank the reviewer for raising this point. For colocalization analyses, it is not possible to apply multiple testing correction across all the pair-wise tests applied with coloc or eCAVIAR as these are Bayesian approaches that output posterior probabilities. The moloc method, however, is a Bayesian statistical framework that integrates GWAS summary data with multiple molecular QTL data and the outputted colocalization posterior probabilities do account for the multiple pairwise comparisons (PMID: 29579179). We included a sentence on this in the Methods section on page 38, lines 918-919.

12) There is no methods section for HiC data or for the CRE. It is difficult to properly evaluate the HiC data and results without its method section. How were CREs defined? How were the CRE identified?

We apologize for this oversight. We have added details on the Hi-C data and CRE and SE data analyzed in our study in the methods section on page 38, lines 920-924.

13) From reading methods section on integration of HiC with other data types, it sounds like the integration is essentially overlap between the different xQTL/M using Genomics Ranges. Is this correct?

Yes, this is correct.

Minor comments

14) QTLs should be changed to QTL throughout the manuscript. The L=Locus (singular) or L=Loci (plural). QTLs is incorrect.

We understand and agree with the reviewer. However, to be able to differentiate between single QTL and multiple QTL, we chose to use the terminology of eQTLs and mQTLs to indicate multiple QTL as done in other widely used eQTL and mQTL studies, including GTEx (PMID: 32913098, 36510025, 35710981, 31076557)

15) How are “target genes” defined in this paper? I would have guessed a target gene was the gene whose expression was mapped to the eQTM, but I also saw “mQTL target genes” (line 215). Does this refer to the hypothetical effect of mQTL on nearby genes? There are also “eQTM target genes” (Line 260). Please define these clearly.

Thank you for helping us clarify the target gene definitions. Target genes for eQTL (also called eGenes) are defined through linear regression analysis of variants in *cis* (± 1 Mb around the TSS)

with gene expression levels. The expression quantitative trait methylation (eQTM) target genes were similarly defined through linear regression analysis of gene expression with DNA methylation levels of CpG sites within ± 1 Mb around the TSS of the target gene. These we called “eQTM target genes”. mQTL target genes, which we called mGenes, were defined based on Illumina’s annotation of Infinium DNA methylation BeadChip probes (Zhou et al., NAR 2017, PMID: 27924034; also see their manifest file: (<https://support.illumina.com/content/dam/illumina-support/documents/downloads/productfiles/methylationepic/infinium-methylationepic-manifest-column-headings.pdf>), which is based on physical proximity of the CpG to gene/s. CpG sites with significant mQTL were mapped to genes based on the CpG’s location relative to the gene’s TSS, gene body or falling with the gene’s 3’ and 5’ UTRs. A CpG could be mapped to more than one gene. We added a sentence clarifying the definition of mQTL target genes (mGenes) (page 10 and 32), eQTM target genes (page 35) and eGenes (pages 9 and 33).

16) The authors define mQTL, eQTL, eQTM in the Results section but not in the Methods section. These should also be defined in the methodology. Others are not clearly defined anywhere, for example “mVariants”.

We apologize for this oversight. We have now defined mQTL, eQTL, eQTM in Methods section and mVariants (genetic variants associated with CpG methylation) (page 10) as well.

17) Sometimes the authors say meQTL instead of mQTL (ex Figure 1F, or in Suppl Figure Legend 2F). Edit to be consistent throughout the manuscript.

We have corrected this in the manuscript, figures, and figure legends.

18) I would suggest to work on the figure labels. For example, the axes on Figure 1B (and multiple similar figures) are confusing. There are two axes, and 3 plots (2 bars and one line), which one goes with which? Similarly, 2F and 2G need axes labels, I see these are chromosomes on the legend, but the axes should be labeled on the figure.

We have now added figure and axis labels for Figure 1B, 2F and 2G.

19) Figure 5A, inset with GE, should “QTL” read “eQTL”?

We apologize for this oversight. We have corrected it to eQTL.

20) Figure 5C, I would advise against using a circle plot. Honestly it is very hard to read it, or see where the lines fall. Even multiple manhattan plots would be better.

We have removed the circle plot in Figure 5C and added a new plot with target genes identified on each chromosome.

Reviewer #3 (Remarks to the Author):

This is a very well-described analysis of human retina tissue integrating genetic, transcriptional, and epigenetic analyses to identify genes and pathways regulated under normal conditions that may also be influenced by advancing age and/or environmental factors.

The authors present a great deal of information to support their analyses. The results were very important, but also very long, and I would find it incredibly helpful to have a bit more interpretation

in the discussion of why each of these analyses were done and how they contribute to the overall findings/message of the paper and to the field.

We added a final conclusion paragraph in the Discussion section (page 27) and have further clarified the interpretation of our results from the various analyses performed in the Discussion (page 25).

I have a few additional questions and suggestions for the authors:

Lines 178-179: "749,158 CpG sites that passed stringent QC criteria (see Methods) using QTLtools42." -- the stringent QC criteria are not found in the methods.

We have now added the stringent QC criteria in the Methods section on page 30, lines 714-718.

Lines 229-232: "To evaluate the tissue-specificity of DNAm and mQTLs, we compared 36,906 methylated CpGs with significant mQTLs and 37,453 independent mQTLs in the retina to those identified in five different tissues: adipose (n=119)45, blood (n=614)46, endometrium (n=66)47, brain frontal cortex (n=526)48, and skeletal muscle (n=282)20." -- Please explain the how and why the comparison tissues of choice were determined.

The tissue comparison was based on the availability of mQTL datasets for different tissues at the time we conducted our analysis. We have clarified this in the main text on page 11, line 233.

Lines 250-252: "Top 10 genotype PCs, known covariates (e.g., AMD grade), and SVs capturing hidden covariates of expression, were included in the linear regression model (see Methods)." -- I do not see any elaboration in the methods of the "known covariates" and would find it helpful to know what those were.

We now listed all known covariates included in the eQTM regression model in the Methods section on page 35, lines 843-844.

Line 275: "on a few smaller chromosomes" -- please explain somewhere in the manuscript why the chromosome size is important for this type of analysis.

We note that chromosomes 16, 17, 19 and 22, though relatively small, are gene rich (PMID: 11181995). We thus inspected the eQTM distribution by chromosome and indeed found high enrichment of eQTMs on a few of the smaller chromosomes. We have added the details about considering the chromosome size in the eQTM analysis on page 24 and clarified this point in the legend of Figure 2.

Given the mention of the effects smoking, exercise, and diet have on epigenetics, and the known influence of these factors on AMD, did the authors have smoking and/or diet data on the retina donors and did they consider utilizing this in analyses in any way?

We agree with the reviewer that considering these factors in our analyses would be interesting. However, since we only have smoking information for the retina donors from the Bethesda dataset, and not for the retina donors from the Cologne and Munich dataset, we were not able to use this information in the analysis. Furthermore, we do not have information on exercise or diet on any of the retina donors from both the datasets.

To appeal to readers who are less familiar with these types of analyses, how they are interpreted, and why they are important, a concise concluding statement/section on the implications of these findings would be helpful.

We thank the reviewer for this useful suggestion. We have added concluding sentences at the end of the Discussion on page 27.

REVIEWERS' COMMENTS

Reviewer #1 (Remarks to the Author):

Authors addressed my concern. No additional comments.

Reviewer #2 (Remarks to the Author):

I would like to thank Advani et al. for addressing the comments from the reviewers, and incorporating clarifying information into the manuscript. I believe publication of the manuscript in its current form would be valuable to ocular research community, and would recommend for publication.

Reviewer #3 (Remarks to the Author):

The authors have sufficiently addressed the reviewer concerns.